# Purinergic regulation of vascular tone in the retrotrapezoid nucleus is specialized to support the drive to breathe

Virginia E Hawkins[1], Ana C Takakura[2], Ashley Trinh[1], Milene R Malheiros-Lima[3], Colin M Cleary[1], Ian C Wenker[1], Todd Dubreuil[1], Elliot M Rodriguez[1], Mark T Nelson[4,5], Thiago S Moreira[3*†], Daniel K Mulkey[1*†]

[1]Department of Physiology and Neurobiology, University of Connecticut, Storrs, United States; [2]Department of Pharmacology, Institute of Biomedical Sciences, University of São Paulo, São Paulo, Brazil; [3]Department of Physiology and Biophysics, Institute of Biomedical Sciences, University of São Paulo, São Paulo, Brazil; [4]Department of Pharmacology, College of Medicine, University of Vermont, Burlington, United States; [5]Institute of Cardiovascular Sciences, University of Manchester, Manchester, United Kingdom

*For correspondence: tmoreira@ icb.usp.br (TSM); daniel.mulkey@ uconn.edu (DKM)

†These authors contributed equally to this work

Competing interests: The authors declare that no competing interests exist.

**Abstract** Cerebral blood flow is highly sensitive to changes in $CO_2/H^+$ where an increase in $CO_2/H^+$ causes vasodilation and increased blood flow. Tissue $CO_2/H^+$ also functions as the main stimulus for breathing by activating chemosensitive neurons that control respiratory output. Considering that $CO_2/H^+$-induced vasodilation would accelerate removal of $CO_2/H^+$ and potentially counteract the drive to breathe, we hypothesize that chemosensitive brain regions have adapted a means of preventing vascular $CO_2/H^+$-reactivity. Here, we show in rat that purinergic signaling, possibly through $P2Y_{2/4}$ receptors, in the retrotrapezoid nucleus (RTN) maintains arteriole tone during high $CO_2/H^+$ and disruption of this mechanism decreases the $CO_2$ ventilatory response. Our discovery that $CO_2/H^+$-dependent regulation of vascular tone in the RTN is the opposite to the rest of the cerebral vascular tree is novel and fundamentally important for understanding how regulation of vascular tone is tailored to support neural function and behavior, in this case the drive to breathe.

## Introduction

Cerebral blood flow is highly sensitive to changes in $CO_2/H^+$. An increase in $CO_2/H^+$ causes vasodilation and increased blood flow, which in turn facilitates removal of excess $CO_2/H^+$. This response, known as vascular $CO_2$ reactivity, serves to match blood flow with tissue metabolic needs (*Ainslie and Duffin, 2009*). Maintaining tight control of brain $CO_2/H^+$ levels is critical, as there is only a narrow range that is conducive to normal neural function. For example, a modest alkalosis of just 0.2 pH units can trigger seizure activity; conversely, a similar degree of acidification can inhibit cortical activity (*Schuchmann et al., 2006*). Tissue $CO_2/H^+$ levels are also regulated by respiratory activity. This is accomplished by specialized subsets of neurons known as respiratory chemoreceptors that are activated by an increase in $CO_2/H^+$ (*Guyenet and Bayliss, 2015*). This information is then relayed to respiratory centers to enhance breathing, and consequently facilitate removal of arterial $CO_2$ in the exhaled breath.

The retrotrapezoid nucleus (RTN) is a region critical for respiratory chemoreception (*Guyenet and Bayliss, 2015*). This region contains a subset of neurons that are intrinsically sensitive to changes in $CO_2/H^+$ (*Mulkey et al., 2004*: *Wang et al., 2013*) and relay responses to further

**eLife digest** We breathe to help us take oxygen into the body and remove carbon dioxide. Our cells use the oxygen to break down food to release energy, and as they do so they produce carbon dioxide as a waste product. Cells release this carbon dioxide back into the bloodstream so that it can be transported to the lungs to be breathed out. Carbon dioxide also makes the blood more acidic; if the blood becomes too acidic, tissues and organs may not work properly.

The brain uses roughly 25% of the oxygen consumed by the body and is particularly sensitive to the levels of gases and acidity in the blood. It has been known for more than a century that increased carbon dioxide causes blood vessels in the brain to widen, allowing the excess carbon dioxide to be carried away quickly. More recent work has shown that increased carbon dioxide also activates neurons called respiratory chemoreceptors. These in turn activate the brain centers that drive breathing, causing us to breathe more rapidly to help us remove surplus carbon dioxide.

But this scenario contains a paradox. If high levels of carbon dioxide cause widening of the blood vessels in the brain regions that contain respiratory chemoreceptors, this should, in theory, wash out that important stimulus, reducing the drive to breathe. So how does the brain prevent this unhelpful response? By studying the brains of adult rats, Hawkins et al. show that different rules apply to the brain centers that control breathing compared to other areas of the brain. In one such region, if the blood becomes too acidic, support cells called astrocytes release chemical signals called purines. This counteracts the tendency of high carbon dioxide levels to widen blood vessels in this region, and instead causes these vessels to become narrower.

This mechanism ensures that local levels of carbon dioxide in respiratory brain centers remain in tune with the demands of local networks, thereby maintaining the drive to breathe. The next challenges are to identify the molecular mechanisms that control the diameter of blood vessels in brain regions containing respiratory chemoreceptors, and to find out whether drugs that modulate these mechanisms have the potential to treat some respiratory conditions.

respiratory control regions, such as the ventral respiratory complex to control breathing rate, inspiratory amplitude, active expiration and airway patency (*Guyenet and Bayliss, 2015*; *Silva et al., 2016*). Disrupting mechanisms by which RTN neurons sense $CO_2/H^+$ abolishes ventilatory responses to $CO_2$ and results in severe apnea (*Kumar et al., 2015*). Interestingly, RTN astrocytes also support chemoreception by providing a $CO_2/H^+$-dependent purinergic drive that enhances activity of chemosensitive neurons (*Gourine et al., 2010*; *Wenker et al., 2012*). This function of RTN astrocytes is unique to the RTN since astrocytes elsewhere do not respond similarly to changes in pH (*Gourine et al., 2010*; *Sobrinho et al., 2014*).

For more than a century, vascular $CO_2$ reactivity has been assumed to be a common feature of the entire cerebrovascular tree (*Ainslie and Duffin, 2009*; *Roy and Sherrington, 1890*). However, if $CO_2/H^+$-induced vasodilation were to occur in chemosensitive regions it would accelerate removal of tissue $CO_2/H^+$ and effectively counter-regulate activity of respiratory chemoreceptors (*Xie et al., 2006*). Therefore, we propose that regulation of vascular tone is specialized to support local neural network function, and specifically that a chemoreceptor region like the RTN has evolved a means of maintaining vascular tone during exposure to high $CO_2/H^+$ in a manner that supports the drive to breathe. Consistent with this, early studies showed that fast breath by breath changes in arteriole $CO_2$ correspond with changes in pH measured at the ventral medullary surface (*Millhorn et al., 1984*), suggesting tissue pH in this region is not highly buffered, possibly because blood vessels in this region do not dilate in response to $CO_2/H^+$. Furthermore, considering that $CO_2/H^+$-evoked ATP release appears to be unique feature of RTN chemoreception (*Gourine et al., 2010*) and since ATP can mediate vasoconstriction in other brain regions (*Kur and Newman, 2014*; *Peppiatt et al., 2006*), we hypothesize that $CO_2/H^+$-evoked ATP release will antagonize $CO_2/H^+$-vasodilation in the RTN, and thus prevent $CO_2/H^+$ washout, further enhancing chemoreceptor function.

Consistent with this hypothesis, we find that arterioles in the RTN and cortex are differentially modulated by purinergic signaling during exposure to high $CO_2/H^+$. Specifically, we show in vitro and in vivo that exposure to $CO_2/H^+$ caused vasoconstriction of RTN arterioles but vasodilation of cortical

arterioles. The $CO_2$/$H^+$-response of RTN arterioles was blocked by bath application of a P2 receptor blocker (pyridoxalphosphate-6-azophenyl-2',4'-disulfonic acid; PPADS) and mimicked by a $P2Y_{2/4}$ receptor agonist (UTPγS) but not a P2X receptor agonist (α,β-mATP), suggesting mechanism(s) underlying this response in the RTN involve purinergic signaling and downstream activation of $P2Y_2$ and/or $P2Y_4$ receptors. To support the possibility that RTN vascular control contributes to respiratory behavior, we show that disruption of purinergic regulation of vascular tone or application of a vasodilator (SNP) to the RTN region decreased the ventilatory response to $CO_2$, whereas application of vasoconstrictors (phenylephrine or U46619) potentiated the central chemoreflex. These results suggest for the first time that regulation of vascular tone in a respiratory chemoreceptor region is specialized to support the drive to breathe.

## Results and discussion

We initially tested our hypothesis using the brain slice preparation optimized for detecting increases or decreases in vascular tone (see Mateials and methods). For these experiments, we targeted arterioles based on previously described criteria (*Mishra et al., 2014*; *Filosa et al., 2004*). Vessel diameter was monitored continuously during exposure to 15% $CO_2$ (pH = 6.9) under baseline conditions and during purinergic receptor blockade with PPADS. Consistent with our hypothesis, we found $CO_2$/$H^+$ differentially regulates arteriole diameter in the RTN depending on the function of purinergic receptors. For example, exposure to $CO_2$/$H^+$ under control conditions resulted in a vasoconstriction of $-4.6 \pm 0.6\%$ (p<0.0001, N = 34 vessels) (*Figure 1C*) (estimated by Poiseuille's law to decrease blood flow by ~20%). Further, we found that $CO_2$/$H^+$-induced constriction of RTN arterioles was retained in the presence of tetrodotoxin (TTX; 0.5 µM) to block neuronal action potentials ($-6.1 \pm 1.6\%$, p=0.0103, N = 6 vessels), thus suggesting glutamatergic $CO_2$/$H^+$-activated neurons (*Mulkey et al., 2004*) are not requisite determinants of this response. Conversely, exposure to $CO_2$/$H^+$ did not cause constriction of RTN arterioles during P2 receptor blockade with 5 µM PPADS ($-0.1 \pm 0.9\%$, p=0.4876, N = 8 vessels) (*Figure 1A–C*). We also tested effects of exogenous ATP to confirm that it functions as vasoconstrictor of RTN arterioles. Indeed, we found that exposure to ATP (100 µM) resulted in a $-5.8 \pm 1.5\%$ constriction (p=0.0018, N = 7 vessels) (*Figure 1D–E*). These results show that purinergic signaling contributes to $CO_2$/$H^+$-dependent control of RTN arterioles.

Purinergic receptors are expressed by a wide variety of cell types including neurons, astrocytes, smooth muscle and endothelial cells; in the context of vascular control, the P2 receptors most commonly implicated in vasoconstriction are several members of the P2X family of ionotropic receptors and metabotropic $P2Y_2$, $P2Y_4$ and $P2Y_6$ (*Burnstock and Ralevic, 2014*). Since the concentration of PPADS used above to block purinergic modulation of RTN arterioles has highest affinities for P2X and $P2Y_2$ and $P2Y_4$ (*Ralevic and Burnstock, 1998*), to identify candidate P2 receptors that help maintain RTN arteriole tone during exposure to $CO_2$/$H^+$, we tested effects of a selective $P2Y_2$ and $P2Y_4$ receptor agonist (UTPγS) (*Lazarowski et al., 1996*) and an agonist with high affinity for P2X receptors (α,β-mATP) (*Burnstock and Kennedy, 1985*). We found that bath application of UTPγS (0.5 µM) mimicked effects of $CO_2$/$H^+$ by decreasing diameter of RTN arterioles ($-3.4 \pm 0.6\%$, p=0.0003, N = 8 vessels), whereas exposure to α,β-mATP (100 µM) minimally affected arteriole tone (p=0.2113, N = 9 vessels) (*Figure 1D–E*). These results suggest that mechanism(s) underlying purinergic-dependent control of RTN arterioles during high $CO_2$/$H^+$ involve activation of Gq-coupled $P2Y_{2/4}$ receptors. To further support this possibility, we performed immunohistochemistry using commercially available $P2Y_2$ and $P2Y_4$ specific antibodies in conjunction with cell-type specific markers for endothelial cells (DyLight 594 Isolectin B4 conjugate; IB4), vascular smooth muscle cells (α-smooth muscle actin; α-SMA), and astrocytes (anti-glial fibrillary acidic protein; GFAP). We detected $P2Y_2$ and $P2Y_4$ immunoreactivity in close proximity to all three cell types associated with RTN arterioles. For example, $P2Y_2$ and $P2Y_4$ labeling appeared as numerous intensely stained puncta near endothelial cells and smooth muscle cells and as smaller more diffuse puncta near astrocytes (*Figure 1F–G*). The expression of these receptors together with our functional evidence suggest $P2Y_{2/4}$ receptors contribute to purinergic-dependent vasoconstriction in the RTN during exposure to $CO_2$/$H^+$.

Considering that ATP and UTP breakdown products are known to affect vascular tone in other brain regions (*Burnstock and Ralevic, 2014*), we also pharmacologically manipulated P1 receptors and ectonucleotidase activity before or during exposure to $CO_2$/$H^+$. We found that application of

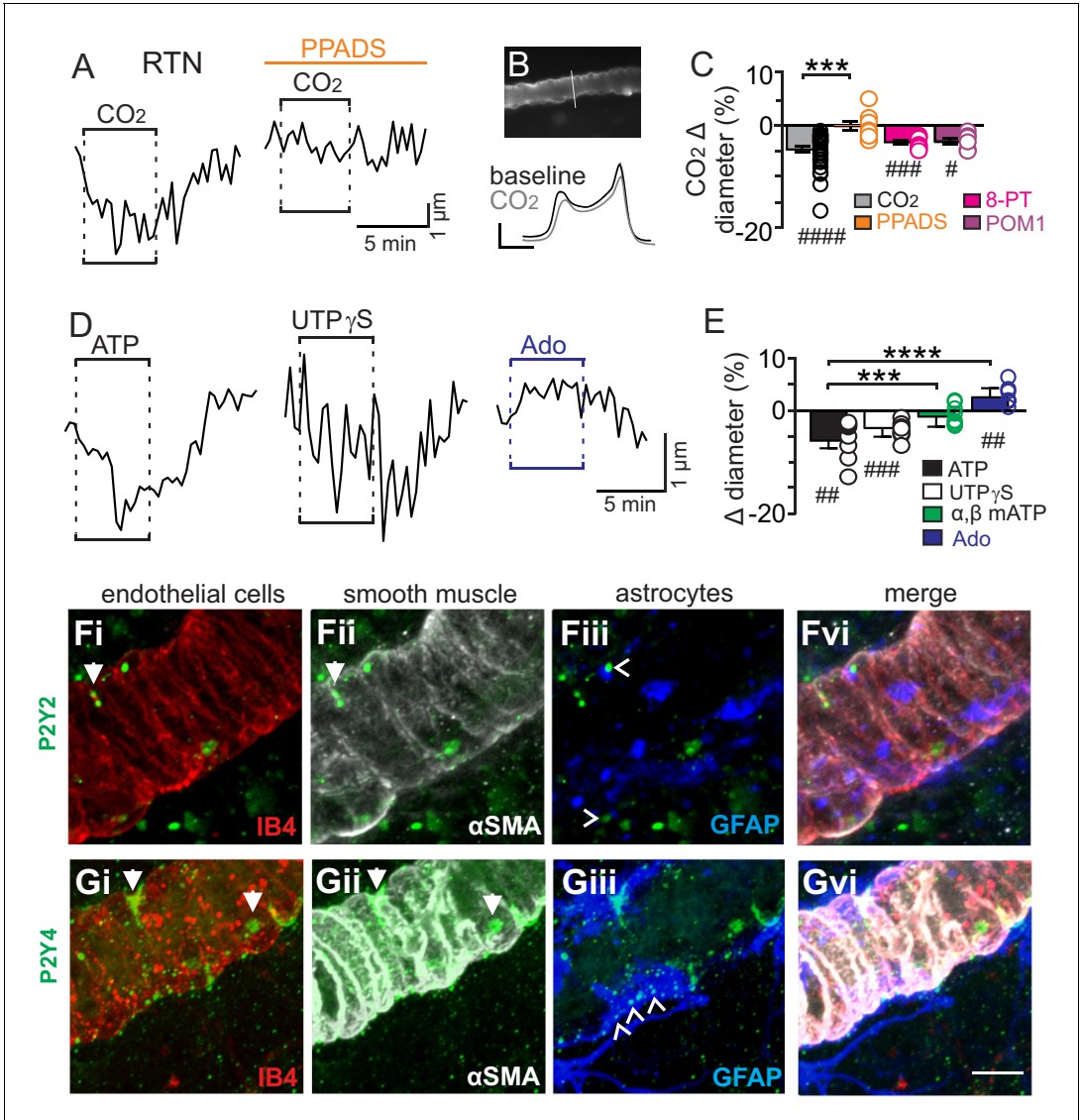

**Figure 1.** $CO_2/H^+$-induced vasoconstriction of RTN arterioles is mediated by a purinergic dependent mechanism involving $P2Y_{2/4}$ receptors. (**A**) trace of an RTN arteriole diameter show that increasing $CO_2$ in the perfusion media from 5% to 15% (balance air, in TTX) caused vasoconstriction under baseline conditions but not in PPADS (5 µM). (**B**) example vessel image under baseline conditions and corresponding fluorescent intensity profile plots also show that exposure to high $CO_2$ decreased vessel diameter. Profile plot scale bars: 2000 a.u., 10 µm. (**C**) summarized results of RTN arteriole responses to $CO_2/H^+$ under baseline conditions (N = 34 vessels) and when P2-receptors were blocked (5 µM PPADS; N = 8 vessels), P1-receptors were blocked (10 µM 8-PT; N = 7 vessels), or ectonucleotidase activity was inhibited (100 µM POM1; N = 5 vessels). (**D**) example diameter traces show RTN arterioles constrict in response to bath application of ATP (100 µM) or the selective $P2Y_{2/4}$ receptor agonist UTPγS (0.5 µM) but dilate when P1 receptors are activated by adenosine (Ado; 1 µM). (**E**) summary data plotted as % diameter change in response to ATP (N = 7 vessels), UTP (N = 8 vessels), α,β-mATP (100 µM, preferential P2X agonist; N = 9 vessels) or adenosine (N = 9 vessels). (**F–G**), immunoreactivity for $P2Y_2$ (**F**) and $P2Y_4$ (**G**) receptors was detected as brightly label puncta near endothelial cells (DyLight 594 Isolectin B4 conjugate; IB4), arteriole smooth muscle (α-smooth muscle actin; αSMA), and astrocytes (glial fibrillary acidic protein; GFAP) associated with arterioles in the RTN (N = 3 animals). Arrows identify receptor labeling close to endothelial or smooth muscle cells and arrowhead identifies receptor labeling of astrocyte processes. Scale bar 10 µM. Hash marks designate a difference in µm from baseline as determined by RM-one-way ANOVA and Fishers LSD test or paired t-test and asterisks identify differences in $CO_2/H^+$-induced % change under baseline conditions vs in the presence of PPADS (**C**) or ATP vs specific agonist-induced % change (**E**) (one-way ANOVA and Fishers LSD test); one symbol = p<0.05, two symbols = p<0.01, three symbols = p<0.001, four symbols = p<0.0001.

adenosine (1 μM) under control conditions caused vasodilation of RTN arterioles (2.6 ± 0.6%; p=0.0027, N = 9 vessels) (*Figure 1D–E*); however, blockade of adenosine receptors with 8-phenyltheophylline (8-PT; 10 μM) had negligible effects on $CO_2/H^+$-induced vasoconstriction (−3.2 ± 0.4%, p=0.0002, N = 7 vessels) (*Figure 1C*). Likewise, incubation in sodium metatungstate (POM 1; 100 μM) to inhibit ectonucleotidase activity also minimally affected the $CO_2/H^+$-vascular response of RTN arterioles (−3.1 ± 0.6%, p=0.0195, N = 5 vessels) (*Figure 1C*). These results suggest that nucleoside metabolites are not essential for $CO_2/H^+$-dependent regulation of vascular tone in the RTN.

Previous evidence (*Gourine et al., 2010*) suggests $CO_2/H^+$-evoked ATP release from RTN astrocytes is mediated by intracellular $Ca^{2+}$. Therefore, in the absence of high $CO_2$, pharmacological activation of RTN astrocytes should trigger arteriole constriction by a purinergic-dependent mechanism. We test this by bath application of t-ACPD (50 μM), an mGluR agonist used to elicit $Ca^{2+}$ elevations in cortical astrocytes (*Filosa et al., 2004*; *Zonta et al., 2003*). Exposure to t-ACPD caused vasoconstriction of RTN arterioles under baseline conditions (−3.5 ± 0.5%, p=0.0007, N = 7) but not in PPADS (−0.6 ± 0.4%, p=0.2163, N = 7 vessels) (*Figure 2A–B,E*). These results are consistent with our hypothesis that purinergic signaling, possibly from $CO_2/H^+$-sensitive RTN astrocytes (*Gourine et al., 2010*), serves to maintain tone of arteriole in the RTN during hypercapnia.

In marked contrast to the RTN, we found that cortical arterioles dilated in response to astrocyte activation by t-ACPD. For example, bath application of t-ACPD (50 μM) dilated cortical arterioles by

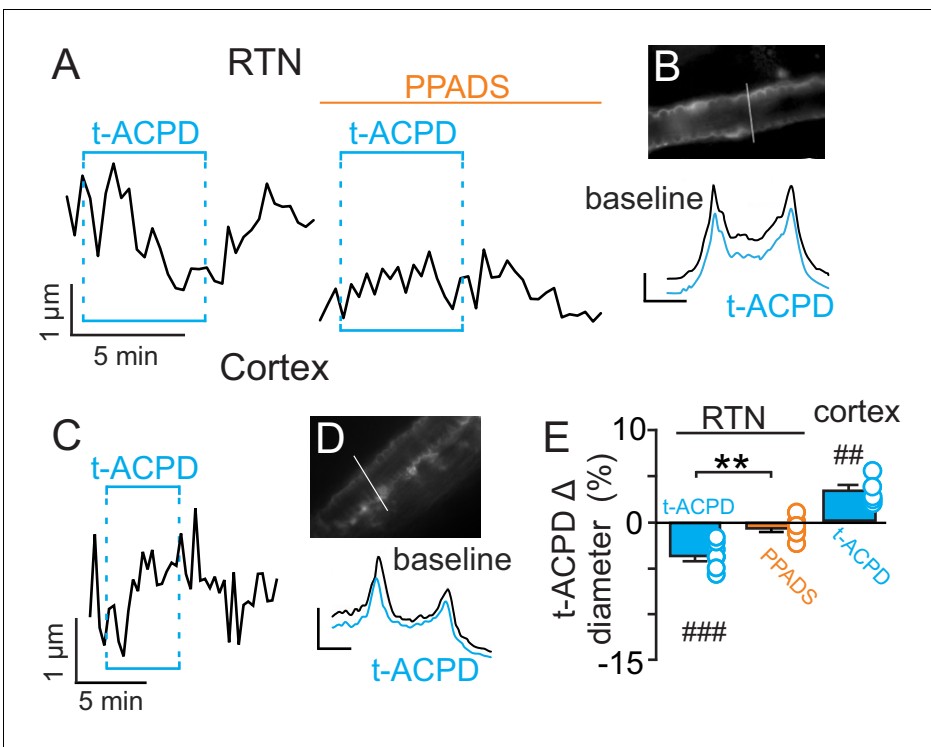

**Figure 2.** t-ACPD-mediated astrocyte activation has opposite effects on arteriole diameter in the RTN and cortex. (A) diameter trace of an RTN arteriole show the response of an RTN arteriole to t-ACPD (50 μM) under baseline conditions and during P2-receptor blockade with PPADS (5 μM). (B) example RTN vessel image under baseline conditions and corresponding fluorescent intensity profile plots also show that exposure to tACPD decreased vessel diameter. (C) diameter trace of an cortical arteriole and corresponding vessel image with example profile plots (D) show that exposure to tACPD (50 μM) increase cortical arteriole diameter. Profile plot scale bars: 2000 a. u., 10 μm. (E) summary from the RTN (N = 7 vessels) and cortex (N = 5 vessels) data show that t-ACPD caused vasoconstriction of RTN arterioles under control conditions but not in the presence of PPADS, suggesting purinergic signaling most likely from astrocytes mediate constriction of arterioles in the RTN. Conversely, in the cortex t-ACPD caused vasodilation. ##, difference in μm from baseline (paired t-test, p<0.01). ###, difference in μm from baseline (RM-one-way ANOVA and Fishers LSD test, p<0.001). **, difference in t-ACPD-induced % change under baseline conditions vs in PPADS (paired t-test, p<0.01).

3.2 ± 0.6% (p=0.0030, N = 5 vessels) (*Figure 2C–E*). This response is consistent with previous cortical studies (*Filosa et al., 2004*; *Zonta et al., 2003*), and suggests that astrocytes in the RTN and cortex have fundamentally different roles in regulation of vascular tone. Also consistent with previous work (*Ainslie and Duffin, 2009*), we confirmed that cortical arterioles dilate in response to $CO_2/H^+$ (5.7 ± 1.1%, p=0.0057, N = 11 vessels) (*Figure 3A–C*). Interestingly, we also found that the $CO_2/H^+$-vascular response of cortical arterioles was reduced to 0.5 ± 0.5% in PPADS (p=0.004, N = 6 vessels) (*Figure 3A–C*), suggesting involvement of endogenous ATP in cortical arteriole $CO_2/H^+$-dilation. As in the RTN, we also found that endothelial cells, smooth muscle and astrocytes associated with cortical arterioles were immunoreactive for $P2Y_2$ and $P2Y_4$ (*Figure 3D–E*), suggesting the differential roles of purinergic signaling in these regions is not due to the presence or absence of $P2Y_2$ and $P2Y_4$. However, since the vascular responses to activation of $P2Y_{2/4}$ can vary depending on which cells express the receptor (*Burnstock and Ralevic, 2014*), it remains possible that differential expression of $P2Y_{2/4}$ by endothelial and smooth muscle may mediate vasodilation in the cortex and constriction in the RTN, respectively. It is also possible that other purinergic receptors contribute to regulation of arteriole tone in these regions. For example, endothelial $P2Y_1$ receptors are known to mediate vasodilation in the cortex (*Burnstock and Ralevic, 2014*). However, we found that in vivo

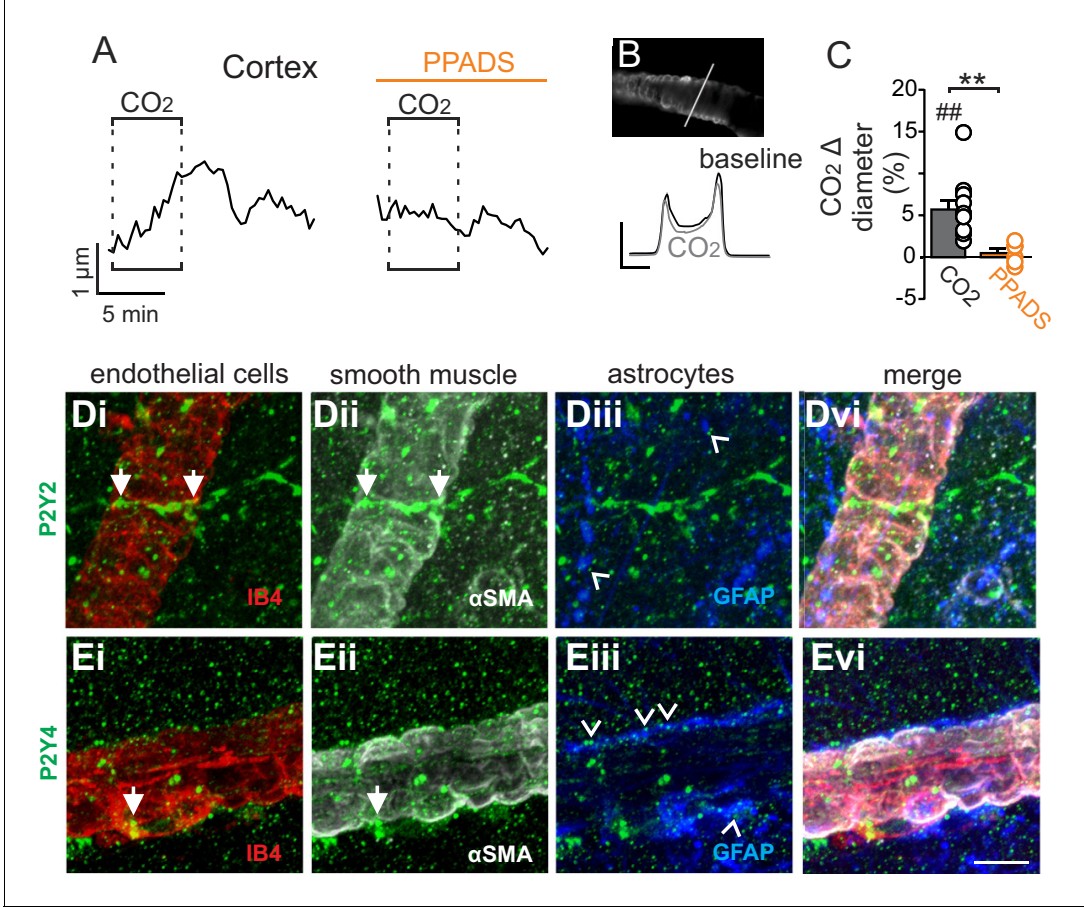

**Figure 3.** Cortical arterioles dilate in response to $CO_2/H^+$. (A) diameter trace of a cortical arteriole with an example vessel image and fluorescence profile plots (B) show that exposure to $CO_2/H^+$ caused vasodilation under baseline conditions and this response was blunted by PPADS (5 µM). Profile plot scale bars: 2000 a.u., 10 µm. (C) summary data show $CO_2/H^+$-induced vasodilation of cortical arterioles under bassline conditions (N = 11 vessels) but not in PPADS (N = 6 vessels). (D–E), immunoreactivity for $P2Y_2$ (D) and $P2Y_4$ (E) receptors was detected as brightly label puncta near endothelial cells (DyLight 594 Isolectin B4 conjugate; IB4), arteriole smooth muscle (α-smooth muscle actin; αSMA), and astrocytes (glial fibrillary acidic protein; GFAP) associated with cortical arterioles (N = 3 animals). Arrows identify receptor labeling close to endothelial or smooth muscle cells and arrowheads identifies receptor labeling of astrocyte processes. Scale bar 10 µM. ##, difference in µm from baseline (paired t test, p<0.01). **, difference in $CO_2/H^+$-induced % change under control conditions vs in PPADS (paired t-test, p<0.01).

application of a selective $P2Y_1$ receptor blocker (MRS2179, 100 µM) had no measurable effect on the $CO_2/H^+$ response of pial vessels in the RTN ($-3.7 \pm 0.8\%$, vs. saline plus $CO_2$: $-4.3 \pm 0.7\%$; p=0.068; N = 5 vessels) or cortex ($4.8 \pm 0.5\%$, vs. saline plus $CO_2$: $4.7 \pm 0.6\%$; p=0.24; N = 5 vessels) (data not shown). Alternatively, arachidonic acid metabolites are also potent regulators of vascular tone (*MacVicar and Newman, 2015*) and recent evidence showed that $CO_2/H^+$-mediated vasodilation in the cortex and hippocampus involved activation of cyclooxygenase-1 and prostaglandin E2 release by astrocytes (*Howarth et al., 2017*). Considering that purinergic signaling can elicit $Ca^{2+}$ responses in astrocytes to facilitate prostaglandin E2 synthesis (*Xu et al., 2003*), it remains possible that purinergic signaling contributes to cortical $CO_2/H^+$ dilation by influencing synthesis and release of prostaglandin E2 by astrocytes. However, currently the cellular and molecular basis of purinergic dilation in the cortex remains unknown.

To determine whether regulation of vascular tone in the RTN impacts respiratory behavior, we pharmacologically manipulated RTN vessels in anesthetized rats while simultaneously measuring systemic blood pressure and diaphragm EMG activity (as a measure of respiratory activity) during exposure to high $CO_2$. We found that localized application of the vasoconstrictors phenylephrine (Phe; 1 µM) or U46619 (1 µM) to the ventral medullary surface (VMS) enhanced the ventilatory response to $CO_2$ by increasing diaphragm electromyogram (EMG) amplitude $15 \pm 2\%$ and $18 \pm 1.8\%$, respectively (*Figure 4A–C*) (p=0.02; N = 6 animals) but with no change in frequency (*Figure 4D*) (p>0.05; N = 6 animals). Also consistent with the possibility that increased blood flow will facilitate removal of tissue $CO_2/H^+$, and thus decrease the stimulus to chemosensitive neurons, we found that VMS application of the vasodilator sodium nitroprusside (SNP; 1 µM) decreased ventilatory response to $CO_2$ by decreasing diaphragm amplitude by $24 \pm 2.6\%$ (*Figure 4A–C*) (p=0.02; N = 6 animals) but with

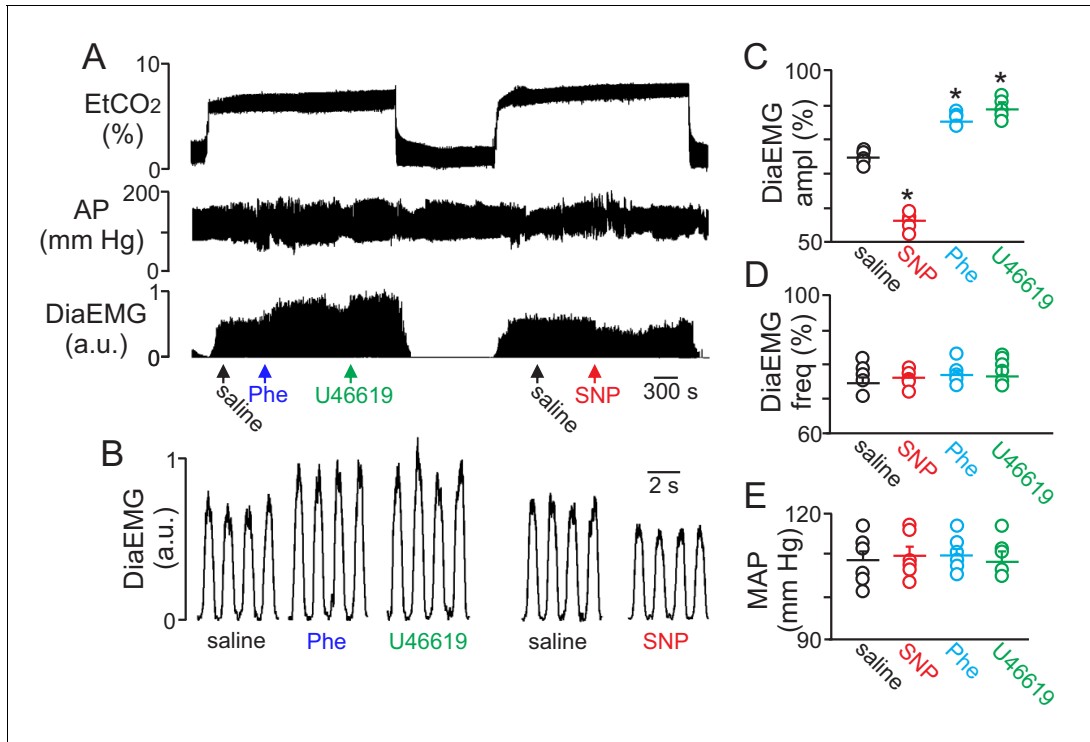

**Figure 4.** Local constriction and dilation of RTN vessels reciprocally modulates the ventilatory response to $CO_2$ *in vivo*. (**A**) end expiratory $CO_2$ ($EtCO_2$), arterial pressure (AP) and diaphragm EMG ($Dia_{EMG}$) traces show that application of vasoconstrictors (phenylephrine, Phe, 1 µM or U46619, 1 µM) or a vasodilator (sodium nitroprusside, SNP, 1 µM) to the RTN increased and decreased the ventilatory response to 7–8% $CO_2$, respectively. (**B**) diaphragm EMG ($Dia_{EMG}$) traces expanded in time show that application of Phe, U46619 or SNP, to the VMS in the region of the RTN increased and decreased the $Dia_{EMG}$ amplitude response to 7–8% $CO_2$. (**C–E**) summary data show effects of saline, SNP, Phe and U46619 applications to the VMS near the RTN on $Dia_{EMG}$ amplitude (N = 6 animals per group) (**C**), $Dia_{EMG}$ frequency (**D**) and mean arterial pressure (MAP) (**E**). *, difference in $CO_2/H^+$-induced % change under control conditions (saline) vs during vasodilation or vasoconstriction (RM-ANOVA followed by Bonferroni multiple-comparison test, p<0.05).

no change in frequency (*Figure 4D*) (p>0.05; N = 6 animals). These treatments had negligible effect on systemic mean arterial pressure (MAP) (Phe: 110 ± 2; SNP: 110 ± 2; U4619: 108 ± 3 vs. saline: 109 ± 2 mmHg; p>0.05) (*Figure 4E*). These results are consistent with the possibility that regulation of vascular tone in the RTN can influence respiratory output. However, we cannot exclude potential direct effects of these drugs on activity of neurons or astrocytes in the region. For example, phenylephrine can directly stimulate activity of chemosensitive RTN neurons (*Kuo et al., 2016*). Therefore, effects of phenylephrine on chemoreception likely involves both vasoconstriction and direct neural activation. It remains to be determined whether U46619 or SNP also affect activity of neurons or astrocytes in the RTN.

To determine whether purinergic signaling regulates $CO_2/H^+$-mediated constriction in vivo, we first measured the diameter of pial vessels on the VMS in the region of the RTN or on the surface of the cortex during exposure to high $CO_2$ under control conditions and when P2 receptor are blocked with PPADS (10 µM). Consistent with our in vitro data, we found that increasing end-expiratory $CO_2$ to 9.5–10%, which corresponds with an arterial pH of 7.2 pH units (*Guyenet et al., 2005*), constricted VMS vessels by −4.5 ± 0.5% (p=0.014, N = 5 animals) (*Figure 5A–B*). However, when P2-receptors are blocked with PPADS (10 µM), increasing inspired $CO_2$ resulted in a vasodilation of 4.3 ± 0.4% (*Figure 5A–B*) (p=0.036; N = 5 animals). This suggests that in the RTN, purinergic-mediated vasoconstriction is working against a background $CO_2/H^+$ dilation, possibly mediated by a cyclooxygenenase/prostroglandine E2-dependent mechanism as described elsewhere in the brain (*Howarth et al., 2017*). Therefore, disruption of $CO_2/H^+$ dilation would be expected to enhance purinergic-dependent vasoconstriction of RTN arterioles, and thus increase baseline breathing and the ventilatory response to $CO_2$. Consistent with this, administration of a cyclooxygenase inhibitor (indomethacin) has been shown to increase baseline breathing and the ventilatory response to $CO_2$ in humans (*Xie et al., 2006*). However, it is also possible that decreasing cerebral blood flow globally by indomethacin treatment or cerebral ischemia (*Chapman et al., 1979*) will cause widespread acidification leading to enhanced activation of multiple chemoreceptor regions including those outside the RTN (*Nattie and Li, 2012*), thus further increasing chemoreceptor drive. It should also be noted that global disruption of cerebrovascular $CO_2/H^+$ reactivity as associated with certain pathological states including heart failure and stroke (*Yonas et al., 1993*; *Howard et al., 2001*) or by systemic administration of indomethacin can increase chemoreceptor gain to the extent that breathing becomes unstable (*Fan et al., 2010*; *Xie et al., 2009*). These results underscore the need to understand how $CO_2/H^+$ regulates vascular tone at other levels of the respiratory circuit.

Also consistent with our slice data, we found that exogenous application of ATP (1 mM) or UTPγS (1 mM) constricted VMS vessels by −5.1 ± 0.6% and −5.0 ± 0.5%, respectively (p=0.001; N = 5 animals) (*Figure 5A*). Conversely, cortical pial vessels dilated in response to an increase in inspired $CO_2$ under control conditions (5 ± 0.5%) and after application of 10 µM PPADS (3.2 ± 0.3%) (*Figure 5C*) (p=0.03; N = 5 animals); however, the $CO_2/H^+$-induced vasodilation of cortical vessels was blunted in the presence of PPADS (p=0.03; N = 5 animals), suggesting endogenous purinergic signaling may facilitate dilation of cortical vessels in response to $CO_2/H^+$. However, we failed to mimic this response by exogenous application of purinergic agonists; exposure to ATP (1 mM) or UTPγS (1 mM) constricted cortical vessels by −4.9 ± 0.7% and −4.1 ± 0.7%, respectively (p=0.024; N = 5 animals). These divergent results are not entirely unexpected since we detected $P2Y_2$ and $P2Y_4$ immunoreactivity on endothelial cells and smooth muscle of cortical arterioles (*Figure 3D–E*), and exogenous application of P2 agonists may activate P2 receptors not necessarily targeted by endogenous purinergic signaling. Future work is required to identify the source of purinergic drive and effector P2 receptors contributing to vascular $CO_2/H^+$-reactivity in the cortex.

In addition, as previously described (*Gourine et al., 2005*), we found that application of PPADS (10 µM) to the RTN blunted the ventilatory response to $CO_2$ both in terms of $Dia_{EMG}$ frequency (82 ± 3, vs. saline: 97 ± 4%) (p=0.042, N = 5) and amplitude (83 ± 6, vs. saline: 100 ± 5%) (p=0.035, N = 5) (*Figure 5D–E*). Considering that RTN manipulations of vascular tone preferentially affect respiratory amplitude (*Figure 4A–D*), whereas application of PPADS to the same region, which likely disrupts both direct excitatory effects of ATP on RTN chemoreceptors (*Wenker et al., 2012*) and indirect effects of ATP on vascular tone (*Figure 5A*), blunts respiratory frequency and amplitude, suggests that purinergic signaling in the RTN might regulate discrete aspects of respiratory output. In the cortex, application of PPADS had no measurable effect on $CO_2$-induced changes in $Dia_{EMG}$ frequency (p=0.33, N = 5 animals) and amplitude (p=0.42, N = 5) (*Figure 5F*). Together with

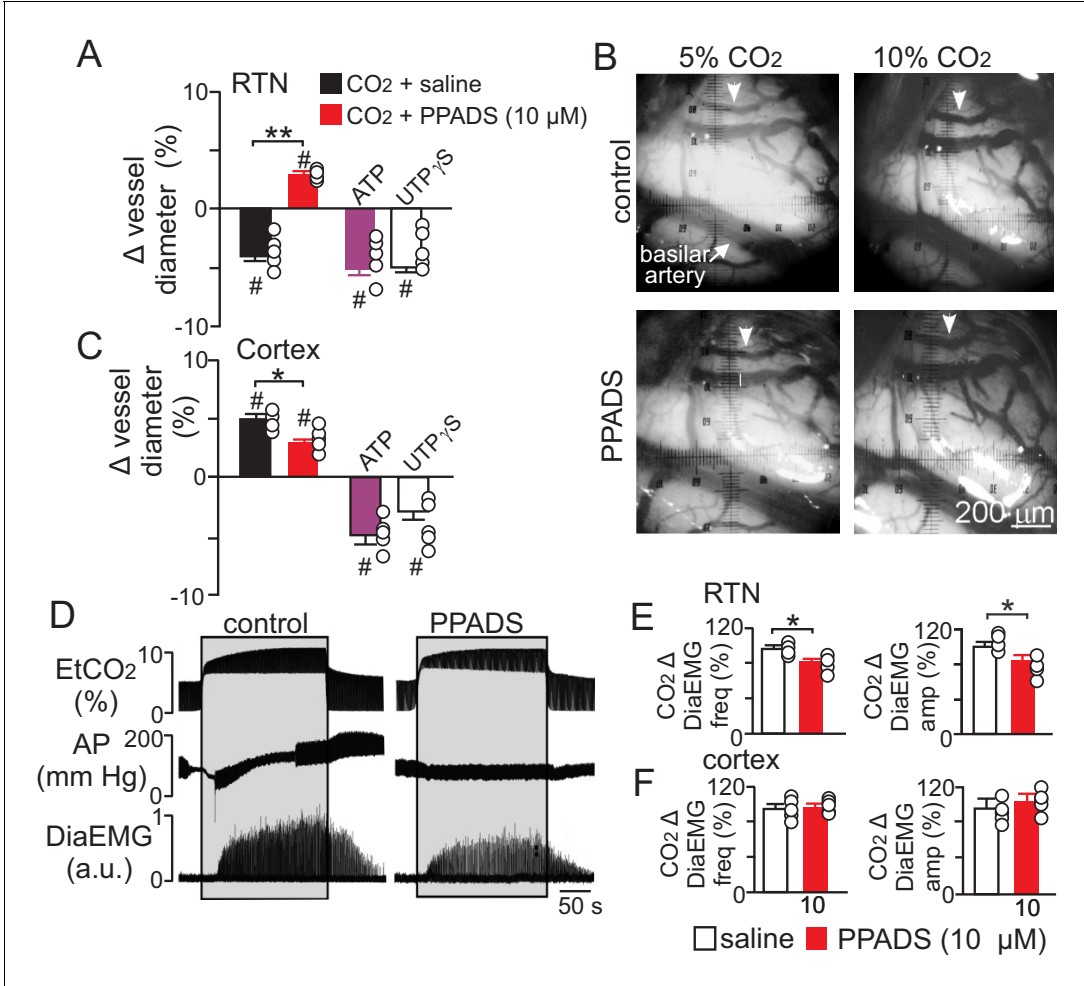

**Figure 5.** Purinergic signaling opposes $CO_2/H^+$-dilation of VMS pial vessels in vivo and contributes to the ventilatory response to $CO_2$. (**A**) summary data plotted as % change in RTN pial vessel diameter in response to an increase in end expiratory $CO_2$ after VMS application of saline (100 nL; N = 5 animals) or PPADS (10 μM, 100 nL; N = 5 animals). Also shown are the vascular responses to VMS application of ATP (1 mM, 100 nL; N = 5) and UTPγS (1 mM, 100 nL; N = 5 animals). (**B**) Photomicrographs (40X) show pial vessel distribution on the VMS, arrow; representative vessel analyzed. (**C**) Summary data shows the response of cortical pial vessels to $CO_2$ after local application of saline or PPADS (10 μM, 100 nL; N = 5 animals). Also shown are vascular responses to exogenous application of ATP (1 mM - 100 nL; N = 5 animals) or UTPγS (1 mM, 100 nL; N = 5). (**D**) End expiratory $CO_2$ (EtCO$_2$), arterial pressure (AP) and diaphragm EMG (Dia$_{EMG}$) traces show that bilateral VMS application of PPADS (10 μM, 100 nL) attenuated the ventilatory response to $CO_2$. (**E**) summary data show $CO_2$-induced changes in Dia$_{EMG}$ frequency and amplitude after bilateral VMS application of saline and PPADS (10 μM; N = 5 animals). (**F**) summary data show that PPADS (10 μM) application to the cortex surface had no measurable effect on $CO_2$-induced changes in DiaEMG frequency and amplitude (N = 5 animals). Hash marks designate a difference in μm from baseline (RM-ANOVA followed by Bonferroni multiple-comparison test, #, $p < 0.05$). Asterisks identify a difference in $CO_2/H^+$–induced % change under control conditions (saline) vs in the presence of PPADS (RM-ANOVA followed by Bonferroni multiple-comparison test, *, $p < 0.05$; **, $p < 0.01$) (panels A and C) or paired t-test (panel E, *, $p < 0.05$).

previous evidence, these findings suggest that purinergic signaling contributes to RTN chemoreception by directly activating RTN neurons (*Gourine et al., 2010*) and indirectly by opposing $CO_2/H^+$-vasodilation.

It should be acknowledged that our study is limited to the use of pharmacological tools that potentially have off-target effects. We have minimized this concern by (i) using low concentrations of PPADS that are reported to be specific for P2 receptor's (*Lorier et al., 2004*); (ii) mimicking $CO_2/H^+$-induced vasoconstriction in vitro and in vivo by exogenous application of ATP and a specific P2Y$_{2/4}$ receptor agonist (UTPγS), but not by a non-specific P2X receptor agonist (α,β-mATP); (iii) confirming that candidate P2Y$_2$ and P2Y$_4$ receptors are expressed in the RTN at the astrocyte-arteriole

interface; and (iv) for in vitro experiments by confirming that $CO_2/H^+$-induced vasoconstriction was retained when neuronal action potentials were blocked with TTX. Therefore, our results suggest that purinergic signaling possibly through $P2Y_{2/4}$ receptors in the RTN provides specialized control of vascular tone by preventing $CO_2/H^+$-induced dilation. Our results also suggest that regulation of vascular tone in the RTN contributes functionally to the ventilatory response to $CO_2$. This is the first evidence to suggest that regulation of vascular tone in a chemoreceptor region contributes to the drive to breathe. This discovery may be of fundamental importance to understanding how regulation of vascular tone impacts neural network function and ultimately behavior.

## Materials and methods

### Brainstem slice preparation

All procedures were performed in accordance with National Institutes of Health and University of Connecticut Animal Care and Use Guidelines. A total of 93 adult Sprague Dawley rats (60–100 days of age) were used for in vitro experiments. Animals were decapitated under isoflurane anesthesia, and transverse brainstem slices (200 μm) were prepared using a vibratome in ice-cold substituted artificial cerebrospinal fluid (aCSF) solution containing (in mm): 130 NaCl, 3 KCl, 2 $MgCl_2$, 2 $CaCl_2$, 1.25 $NaH_2PO_4$, 26 $NaHCO_3$, and 10 glucose, with 0.4 mM L-ascorbic acid added (all Sigma-Aldrich). Slices were incubated for ~30 min at 37°C and subsequently at room temperature in aCSF. Prior to imaging slices were incubated for 1 hr with 10 μg/ml TRITC-lectin conjugate (Sigma-Aldrich, St. Louis, MO) or 6 μg/ml DyLight 594 Isolectin B4 conjugate (Vector Labortories) to label endothelial cells as previously described (*Mishra et al., 2014*). Slicing solutions were equilibrated with a high oxygen carbogen gas (95% $O_2$-5% $CO_2$) (*Mulkey et al., 2001*).

### Imaging of arterioles in brainstem slices

An individual slice containing the RTN was transferred to a recording chamber mounted on a fixed-stage microscope (Zeiss Axioskop FS) and perfused continuously (~2 ml $min^{-1}$) with aCSF bubbled with 5% $CO_2$, 21% $O_2$ (balance $N_2$; pHo ~7.35; $37^0$ C) (*Gordon et al., 2008*). Hypercapnic solution was made by equilibrating aCSF with 15% $CO_2$, 21% $O_2$ (balance $N_2$; pHo ~6.90; $37^0$ C). Arterioles were identified as previously described (*Mishra et al., 2014*; *Filosa et al., 2004*) by the following criteria: clear evidence of vasomotion under IR-DIC, bulky fluorescent labeling and a thick layer of smooth muscle surrounding the vessel lumen. Vessels that appeared collapsed and unhealthy were excluded, as were those with little fluorescence staining and thin walls, indicative of a lack of smooth muscle (*Mishra et al., 2014*). All arterioles selected for analysis had a resting luminal diameter of between 8–45 μm; RTN vessels were located within 200 μm of the ventral surface and below the caudal end of the facial motor nucleus and cortical vessels were located in layers 1–3.

For an experiment, fluorescent images were acquired at a rate of 1 frame/20 s using a x40 water objective, a Clara CCD Andor camera and NIS Elements software. To induce a partially constricted state we bath applied a thromboxane A2 receptor agonist (U46619; 125 nM; Sigma-Aldrich, St. Louis, MO). At this concentration, U46619 has been shown to decrease vessel diameter by 20–30% under similar experimental conditions, thus allowing assessment of both vasodilation and vasoconstriction (*Filosa et al., 2004*; *Girouard et al., 2010*). In the continued presence of U-46619, we then characterized the effects of hypercapnia, ATP (100 μM; Sigma-Aldrich, St. Louis, MO), α,β-methyleneATP (100 μM), UTPγS (0.5 μM), and adenosine (1 μM), or the mGluR agonist t-ACPD ((±)−1-amino-cyclopentane-*trans*-1,3-dicarboxylic acid; 50 μM) alone or in the presence of P2-recepetor blocker PPADS (5 μM; Tocris Bioscience, Minneapolis, MN), the P1 receptor antagonist 8-Phenyltheophylline (8-PT; 10 μM; Sigma) or the Ecto-NTPDase antagonist sodium metatungstate (POM 1; 100 μM; Tocris). In a subset of experiments we also tested $CO_2$, ATP and PPADS in the presence of TTX (0.5 μM; Alomone Laboratories). As previously described (*Girouard et al., 2010*), at the end of each experiment we assessed arteriole viability by inducing a constriction with a high $K^+$ solution (60 mM) and then maximal dilation with a $Ca^{2+}$ free solution containing EGTA (5 mM), papaverine (200 μM, a phosphodiesterase inhibitor) and diltiazem (50 μM, to block L-type Ca channels). Vessels from both regions of interest (RTN and cortex) show similar responses to these conditions, and vessels that did not respond were excluded from analysis.

## Immunohistochemistry

Rat brain slices were prepared from three rats and labelled with DyLight 594 Isolectin B4 conjugate as above followed by immersion fixation overnight in 1% paraformaldehyde in pH 7.4 PBS at 4°C. Excess fixative was removed by three washes in PBS, and prior to antibody incubations, tissue sections were treated to unmask epitopes with 0.2 mg/ml pepsin (Sigma-Aldrich, St. Louis, MO) in 0.2 M HCl for 10 mins at 37°C (*Corteen et al., 2011*) followed by three washes in PBS for $P2Y_4$ labelling only. A blocking stage was then performed by incubating tissue in 10% normal horse serum in PBS with 10% Triton X-100 (Sigma-Aldrich, St. Louis, MO) for 1 hr at room temperature (RT). Sections were then incubated overnight at RT with primary antibodies diluted in blocking solution as follows: 1:200 rabbit anti-$P2Y_2$ (RRID: AB_2040078) or $P2Y_4$ receptor (RRID: AB_2040080) (Alomone Labs, Alomone Labs, Jerusalem Israel), 1:200 chicken anti-glial fibrillary acidic protein (RRID: AB_177521) (Chemicon) and 1:500 mouse anti-$\alpha$-smooth muscle actin (RRID: AB_262054) (Sigma-Aldrich, St. Louis, MO). After washes in PBS, tissues were incubated for 1 hr at RT with the appropriate secondary antibodies raised in donkey, conjugated with [488]DyLight 1:800 (RRID: AB_2492289), [405]DyLight 1:200 (RRID: AB_2340373) or Cy5 1:500 (RRID: AB_2340820) (Jackson Immunoresearch Laboratories). Sections were washed in PBS again before being mounted with Vectasheild (Vector-Labs). Images were acquired using a Nikon A1R confocal microscope (Nikon Instruments), with minimal background staining observed in the control reactions where primary antibodies were omitted or P2 receptor antibodies were pre-absorbed with control antigen prior to exposure to tissues. For confocal photomicrographs, two-dimensional flattened images of the projected z-stacks are presented.

## In vivo preparation

Animal use was in accordance with guidelines approved by the University of São Paulo Animal Care and Use Committee. A total of 21 adult male Wistar rats (60–90 days of age; 270–310 g) were used for in vivo experiments. General anesthesia was induced with 5% halothane in 100% $O_2$. A tracheostomy was made and the halothane concentration was reduced to 1.4–1.5% until the end of surgery. The femoral artery was cannulated (polyethylene tubing, 0.6 mm o.d., 0.3 mm i.d., Scientific Commodities) for measurement of arterial pressure (AP). The femoral vein was cannulated for administration of fluids and drugs. Rats were placed supine onto a stereotaxic apparatus (Type 1760; Harvard Apparatus) on a heating pad and core body temperature was maintained at a minimum of 36.5°C via a thermocouple. The trachea was cannulated. Respiratory flow was monitored via a flow head connected to a transducer (GM Instruments) and $CO_2$ via a capnograph (CWE, Inc,) connected to the tracheal tube. Paired EMG wire electrodes (AM-System) were inserted into the diaphragm muscle to record respiratory-related activity. After the anterior neck muscles were removed, a basio-occipital craniotomy exposed the ventral medullary surface, and the dura was resected. After bilateral vagotomy, the exposed tissue around the neck and the mylohyoid muscle was covered with mineral oil to prevent drying. Baseline parameters were allowed to stabilize for 30 min prior to recording.

## In vivo recordings of physiological variables

Mean arterial pressure (MAP), diaphragm muscle activity ($Dia_{EMG}$) and end-expiratory $CO_2$ ($etCO_2$) were digitized with a micro1401 (Cambridge Electronic Design), stored on a computer, and processed off-line with version 6 of Spike 2 software (Cambridge Electronic Design). Integrated diaphragm activity ($\int Dia_{EMG}$) was collected after rectifying and smoothing ($\tau = 0.03$) the original signal, which was acquired with a 300–3000 Hz bandpass filter. Noise was subtracted from the recordings prior to performing any calculations of evoked changes in $Dia_{EMG}$. A direct physiological comparison of the absolute level of $Dia_{EMG}$ activity across nerves is not possible because of non-physiological factors (e.g., muscle electrode contact, size of muscle bundle) and the ambiguity in interpreting how a given increase in voltage in one EMG relates to an increase in voltage in another EMG. Thus, muscle activity was defined according to its baseline physiological state, just prior to their activation. The baseline activity was normalized to 100%, and the percent change was used to compare the magnitude of increases or decreases across muscle from those physiological baselines.

## In vivo experimental protocol

Each in vivo experiment began by testing responses to hypercapnia by adding pure $CO_2$ to the breathing air supplied by artificial ventilation. In each rat, the addition of $CO_2$ was monitored to reach a maximum end-expiratory $CO_2$ between 9.5% and 10%, which corresponds with an estimated arterial pH of 7.2 based on the following equation: $pHa = 7.955 - 0.7215 \times log10 (ETCO_2)$ (*Guyenet et al., 2005*). This maximum end-expiratory $CO_2$ was maintained for 5 min and then replaced by 100% $O_2$.

To determine whether local regulation of vascular tone in the region of the RTN contributes to the $CO_2/H^+$-dependent drive to breathe, we made bilateral injections of saline, phenylephrine (Phe, 1 µM), U46619 (1 µM) or sodium nitroprusside (SNP, 1 µM) while monitoring $Dia_{EMG}$ amplitude and frequency. These drugs were diluted to 1 µM with sterile saline (pH 7.4) and applied using single-barrel glass pipettes (tip diameter of 25 µm) connected to a pressure injector (Picospritzer III, Parker Hannifin Corp, Cleveland, OH). For each injection, we delivered a volume of 100 nl over a period of 5s. Injections in the VMS region were placed 1.9 mm lateral from the basilar artery, 0.9 mm rostral from the most rostral hypoglossal nerve rootlet, and at the VMS. The second injection was made 1–2 min later at the same level on the contralateral side. In separate series of experiments saline, ATP (1 mM), UTPγS (1 mM) or PPADS (10 µM) were applied similarly to the VMS to test the effect of P2-blockade on vascular $CO_2/H^+$ reactivity and the ventilatory response to $CO_2$. A cranial optical window was prepared using standard protocols previously described (*Kim et al., 2015*). Briefly, a dental drill (Midwest Stylus Mini 540S, Dentsply International) was used to thin a circumference of a 4–5 mm-diameter circular region of the skull over somatosensory cortex (stereotaxic coordinates: AP: −1.8 mm from bregma and ML: 2.8 mm lateral to the midline). For the VMS, the anterior neck muscles were removed, a basio-occipital craniotomy exposed the ventral medullary surface, and the dura was resected. Pial vessels in the VMS had an average and were located 1.9 mm lateral from the basilar artery and 0.9 mm rostral to the most rostral portion of the hypoglossal nerve rootlet. Both thinned bone were lifted with forceps. The surface of the cortex or the VMS were cleaned with buffer containing (in mmol/L) the following: 135 NaCl, 5.4 KCl, 1 $MgCl_2$, 1.8 $CaCl_2$, and 5 HEPES, pH 7.3., and a chamber (home-made 1.1-cm-diameter plastic ring was glued with dental acrylic cement attached to a baseplate). The chamber was sealed with a circular glass coverslip (#1943–00005, Bellco). The baseplate was affixed to the Digital Camera (Sony, DCR-DVD3-5) and a light microscope was used for vessel imaging (x40 magnification).

## Image analysis

Vessel diameter was determined offline using ImageJ. For in vitro experiments, stack registration and selection of linear regions of interest (ROIs) (three regions per arteriole) was carried out. Linear ROI's were used to create a fluorescent intensity profile plot as described previously (*Mishra et al., 2014*) and a macro (available at https://github.com/omsai/blood-vessel-diameter [*Nanda, 2017*]; copy archived at https://github.com/elifesciences-publications/blood-vessel-diameter).was used to determine the peak-peak distance as a measure of vessel diameter for each frame. In most cases, we also confirmed vessel diameter by manually measuring at least one point per frame. In vivo data was also analyzed using three linear ROI's drawn perpendicular to the vessel in each image and the Diameter plug-in function in ImageJ was used to calculate changes in diameter (*Kim et al., 2015*; *Fischer et al., 2010*).

## Data analysis and statistics

All in vitro images were calibrated and pixel distances converted to diameter (µM) and these values were used for analysis of $CO_2/H^+$ or agonist-induced changes in vessel diameter from baseline by RM-one-way ANOVA and Fishers LSD test or paired t-test. Hash marks were used to identity differences from baseline (vasoconstriction or vasodilation). Mean percent changes in vessel diameter was used to compare between agonist responses or $CO_2/H^+$ responses under control conditions vs during purinergic receptor blockade or in the presence of an ectonucleotidase inhibitor by one-way ANOVA and Fishers LSD test or t-test. Asterisks were used to identify differences in % change in vessel diameter. For in vivo experiments, respiratory muscle activity was calculated as the mean amplitude of the integrated $Dia_{EMG}$ over 20 respiratory cycles. To obtain control values, the 20 cycles preceding each experimental manipulation for all parameters were averaged. Under hypercapnic

conditions, measurements from the 20 cycles preceding stimulus cessation were averaged. Respiratory frequency (fR) was (1/(inspiratory time + expiratory time). Differences in the ventilatory response to $CO_2$ were determined using either paired t-test or one-way analysis of variance (ANOVA) followed by the Bonferroni multiple-comparisons as appropriate. Power analysis was used to determine sample size, all data sets were tested for normality using Shapiro-Wilk test. All data values are expressed as means ± SEM and specific statistical test and relevant p values are reported in the text and figure legends.

## Acknowledgements

We thank David Attwell and Anusha Mishra (University College London), Catherine Hall (University of Sussex) and Pariksheet Nanda (University of Connecticut) for assistance with vessel image analysis. This work was supported by funds from the National Institutes of Health Grants HL104101 (DKM), HL126381 (VEH), and P01-HL-095488, R01-HL-121706, R37-DK-053832 and R01-HL-131181 (MTN). Additional support comes from the Totman Medical Research Trust (MTN), Fondation Leducq (MTN), EC Horizon 2020 (MTN), Connecticut Department of Public Health Grant 150263 (DKM), and public funding from the São Paulo Research Foundation (FAPESP) Grants 2014/22406-1 (ACT), 2016/22069–0 (TSM), 2015/23376–1 (TSM) and FAPESP fellowship 2014/07698-6 (MRML); Conselho Nacional de Desenvolvimento Científico e Tecnológico (CNPq); grant: 471263/2013–3 (ACT) and 471283/2012–6 (TSM). CNPq fellowship 301651/2013–2 (ACT) and 301904/2015–4 (TSM).

## Additional information

### Funding

| Funder | Grant reference number | Author |
|---|---|---|
| National Institutes of Health | HL104101 | Daniel K Mulkey |
| Totman Medical Research Trust | | Mark T Nelson |
| Fondation Leducq | | Mark T Nelson |
| European Commission | Horizon 2020 | Mark T Nelson |
| Connecticut department of public health | 150263 | Daniel K Mulkey |
| Fundação de Amparo à Pesquisa do Estado de São Paulo | 2014/22406-1 | Ana C Takakura |
| Conselho Nacional de Desenvolvimento Científico e Tecnológico | 471263/2013-3 | Ana C Takakura |
| National Institutes of Health | HL126381 | Virginia E Hawkins |
| National Institutes of Health | HL095488 | Mark T Nelson |
| National Institutes of Health | DK053832 | Mark T Nelson |
| National Institutes of Health | HL131181 | Mark T Nelson |
| Fundação de Amparo à Pesquisa do Estado de São Paulo | 2016/22069-0 | Thiago S Moreira |
| Fundação de Amparo à Pesquisa do Estado de São Paulo | 2015/23376-1 | Thiago S Moreira |
| Fundação de Amparo à Pesquisa do Estado de São Paulo | 2014/07698-6 | Milene R Malheiros-Lima |
| Conselho Nacional de Desenvolvimento Científico e Tecnológico | 471283/2012-6 | Thiago S Moreira |
| Conselho Nacional de Desenvolvimento Científico e Tecnológico | 30165½013-2 | Ana C Takakura |

| Conselho Nacional de Desen-volvimento Científico e Tecno-lógico | 301904/2015-4 | Thiago S Moreira |

The funders had no role in study design, data collection and interpretation, or the decision to submit the work for publication.

## Author contributions

VEH, Data curation, Formal analysis, Supervision, Project administration, Writing—review and editing; ACT, Conceptualization, Data curation, Formal analysis, Funding acquisition, Investigation, Project administration, Writing—review and editing; AT, TD, Data curation, Formal analysis, Methodology, Writing—review and editing; MRM-L, CMC, EMR, Data curation, Formal analysis, Investigation, Writing—review and editing; ICW, Conceptualization, Data curation, Formal analysis, Investigation, Writing—review and editing; MTN, Conceptualization, Supervision, Funding acquisition, Investigation, Methodology, Writing—review and editing; TSM, Conceptualization, Data curation, Formal analysis, Supervision, Funding acquisition, Investigation, Project administration, Writing—review and editing; DKM, Conceptualization, Supervision, Funding acquisition, Writing—original draft, Project administration, Writing—review and editing

## Author ORCIDs

Virginia E Hawkins, http://orcid.org/0000-0001-9505-6776
Daniel K Mulkey, http://orcid.org/0000-0002-7040-3927

## Ethics

Animal experimentation: All in vitro procedures were performed in accordance with National Institutes of Health and University of Connecticut Animal Care and Use Guidelines (protocol # A16-034). All in vivo procedures were performed in accordance with guidelines approved by the University of São Paulo Animal Care and Use Committee (protocol # 112/2015).

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
