## [Decision Letter]

[Editors’ note: a previous version of this study was rejected after peer review, but the authors submitted for reconsideration. The first decision letter after peer review is shown below.]

Thank you for submitting your work entitled "Purinergic signaling provides specialized control of vascular tone to support the drive to breathe" for consideration by *eLife*. Your article has been favorably evaluated by a Senior Editor) and three reviewers, one of whom, Jan-Marino Ramirez (Reviewer #1), is a member of our Board of Reviewing Editors.

Our decision has been reached after consultation between the reviewers. Based on these discussions and the individual reviews below, we regret to inform you that your work will not be considered further for publication in *eLife*.

The reviewers saw the importance of your significance, but felt that the pharmacology was not sufficiently convincing to support your conclusions. They suggest testing your hypothesis with additional drugs that are more specific.

*Reviewer #1:*

The study by Hawkins and collaborators demonstrates that the effect of CO_2_/H^+^ on the vascular tone in the RTN, the presumed site for central chemoreception, is significantly different from the vascular control in the cortex.

This is interesting for two main reasons:

A) It is the first demonstration that CO_2_ does not cause a uniform vasodilation throughout the nervous system, but that it differentially controls an area that senses CO_2_.

B) The study is also interesting as it links chemosensation to concurrent changes in vascular tone. This is an interesting and novel concept that will trigger further studies aimed at better defining this interaction.

In this study, the role of purinergic modulation of vascular tone is primarily explored by pharmacological manipulations, as is the in vivo demonstration. The use of phenylephrine for example will affect vascular tone, but this manipulation might also have other indirect effects. Given the novelty of this observation, I am not too concerned that most mechanistic insights are based on pharmacological manipulations. However, being primarily a pharmacological study the authors cannot definitely demonstrate e.g. that it is the asctrocytic ATP which antagonizes the CO_2_/H^+^ vasodilation.

*Reviewer #2:*

This is a potentially interesting paper that makes the case that ATP signaling mediates a specialized form of signaling in the brain areas that control breathing. Overall, I found this finding to be a useful contribution, but was not convinced that it was sufficiently novel for a general audience. In some ways, it seems appropriate for a specialized audience of researchers working on breathing.

My major concern for the paper is the use of 100 μm suramin and PPADS to implicate purinergic signaling. At these concentrations, neither drug is specific and their ability to block many other receptors is well described in published work. Moreover, much better blockers are available that target distinct ATP receptors. My suggestion is to repeat the key experiments with more selective antagonists and/or to repeat the experiments with suramin and PPADS at 3-10 uM, at which doses they are considered more selective for ATP receptors over receptors for other neurotransmitters. Without a more complete evaluation of ATP's involvement, the central message of this paper appears to be on weak foundations.

Finally, some of the traces in Figure 1 showing changes in blood vessel diameter over time were not particularly clear or convincing (e.g. Figure 1D1, A2, E2). Perhaps bolster these experiments with another in vitro approach to be sure the metrics used are reliable.

*Reviewer #3:*

This is manuscript begins to address the intriguing idea that vascular tone in the chemosensitive area, the retrotrapezoid nucleus (RTN) responds differently to increasing CO_2_/H^+^, and thus, likely contributing to the chemosensitivity response. The authors used both in vitro and in vivo preparations to examine changes in arteriole diameter on the ventrolateral medullary surface during increased CO_2_/H^+^ and after blocking purinergic receptors with suramin or PPADS. Cumulatively, the authors suggest that their results demonstrate that the arterioles near the RTN respond to ATP released from astrocytes to prevent vasodilation in response to increasing CO_2_/H^+^. This is a very interesting and important idea; however, two major issues fail to be addressed in this manuscript. Firstly, the authors also fail to address why different purinergic receptor antagonists are used in the in vitro (suramin) and in vivo (PPADS) experiments. Secondly, the authors fail to acknowledge that suramin, and to a lesser extent PPADS, are associated with non-specific effects, particularly at glutamate receptors in the concentrations used here (for example, see Gu et al., 1998, Neuroscience Letters; Motin and Bennett, 1995, Br. J. Pharmacol.; Nakazawa et al., 1995, Naunyn Schmiedebergs Arch. Pharmacol; Lambrecht, 2000, Naunyn Schmiedebergs Arch. Pharmacol). These two issues at a minimum need to be addressed in the Discussion. In addition, the authors need to address how these non-specific actions could alter their conclusions. Lastly, the authors state clearly in the Introduction that they hypothesize "CO_2_/H^+^-evoked astrocyte ATP release will antagonize CO_2_/H^+^-vasodilation in the RTN…"; however, this is not actually the hypothesis that the authors test since they do not directly test the involvement of astrocytes. While this is a logical conclusion based on previous work and in conjunction with the studies tested here, the hypothesis stated needs to accurately describe the experiments herein. In other words, the hypothesis tested is that ATP released during increased CO_2_/H^+^ does not cause vasodilation in the RTN. Despite these issues, the data presented is very interesting and has the potential to foster additional studies to examine the interaction between astrocytes and the vasculature in this area, similar to work in the cortex.

[Editors’ note: a second version of this study was rejected after peer review, but the authors submitted for reconsideration. The decision letter after peer review is shown below.]

Thank you for submitting your work entitled "Purinergic signaling provides specialized control of vascular tone to support the drive to breathe" for consideration by *eLife*. Your article has been favorably evaluated by a Senior Editor and three reviewers, one of whom, Jan-Marino Ramirez (Reviewer #1), is a member of our Board of Reviewing Editors.

Our decision has been reached after consultation between the reviewers. Based on these discussions and the individual reviews below, we regret to inform you that your work will not be considered further for publication in *eLife*.

The reviewers continue to believe that the study is potentially of general interest, and that your finding is significant. But, the reviewers felt that the authors continue to overstate their findings as detailed in the reviews.

Most importantly, the reviewers remain unconvinced that the study demonstrates the purinergic modulation. The study still reports experiments in which the drugs are used at concentrations that are non-specific. Moreover, it is not always clear what concentrations were used. One reviewer suggests additional experiments with P2X and P2Y receptor agonists and antagonists that would more specifically test the proposed mechanism. As also suggested, the addition of immunohistochemistry would clearly strengthen the study by demonstrating that there is indeed the expression of the relevant receptors. Inhibitors of ectoATPases could be tried to explore endogenous ATP involvement, and apyrase could be used to degrade ATP. Although, one reviewer suggested the use of knock-out mice, there was not consensus that this is really necessary for the main conclusion.

*Reviewer #1:*

This revised manuscript has greatly improved and the authors provide various convincing arguments why this study is interesting for a general readership. I agree with the authors that the local control of vascular tone has general implication and is not only relevant for those interested in respiration. Moreover, the authors have been very responsive and added various new experiments that strengthen their conclusions.

The authors used for example an additional vasoconstrictor that has been frequently used, thus addressing a previous concern, and strengthening the conclusion that the RTN is involved in regulating vascular tone. The use of PPADS was suggested by the reviewers and the authors added new experiments to strengthen their pharmacological characterization of purinergic signaling, which is consistent with the findings obtained with the exogenous application of ATP.

Another strength is the combination of in vitro and in vivo approaches. The authors added new figures and re-arranged figures to enhance the clarity of the manuscript. Thus, the authors have addressed all my previous concerns.

*Reviewer #2:*

The manuscript by Hawkins et al. is vastly improved by the revisions. The authors addressed the concerns brought forth by the reviewers. And this reviewer remains interested in the results; however, a few lingering issues remain that decrease the clarity of the manuscript. Mainly, some of the results of the current experiments continue to be overstated (see below). As discussed in the previous review, while these conclusions are plausible given previously published results, the results from the present experiments do not directly lead to such conclusions. Wording adjustments are necessary to make clear what is actually shown in the current paper. Secondly, inconsistencies between the data reported in the text of the results and the figures (specifically the bar graphs) is concerning (see below for details). These discrepancies make it challenging to accurately evaluate the manuscript. Overall, this manuscript remains of interest to this reviewer with a few more revisions.

1) Concluding sentence overstatements

"Together, these results suggest that astrocytes and purinergic signaling have fundamentally different roles in the regulation of vascular tone in the RTN and cortex, with the RTN being specialized to support chemoreceptor function." The reviewer would agree with the conclusion about purinergic signaling, but these data do not directly address involvement of astrocytes. The authors measure arterioles, not astrocyte activity.

"These results provide clear evidence that increasing or decreasing vascular tone in the RTN will influence the response to CO_2_. Therefore, we conclude that regulation of vascular tone in the RTN contributes to the chemoreflex." The conclusion for these results is more appropriately worded in the response to reviewers. Since the authors did not verify vascular tone in the experiments related to this conclusion, this conclusion is overstated for what is shown. Despite the addition of the U46619 data, the data does not indicate these drugs ONLY altered blood vessels. Based on the data presented, they cannot rule out diffusion of the drugs and altering astrocytes or neurons in VLM.

2) Data inconsistencies

Figure 1 and Results. Overall, it is unclear which statistical tests were run and which data were involved in which tests. The Methods indicate both t-tests and ANOVAs were run, but it is not clear when each was run or how data were grouped to determine significance.

3) It is unclear what "control" is for Figure 1. Is control the response to CO_2_ for all data in Figure 1? In specific, is "con" referring to the CO_2_ response or the t-ACPD in Figure 1E1? The same question for Figure 1F2. If in both cases, "con" refers to the CO_2_ response and if that is included when running the statistics, why does this bar not appear in the summary data?

4) Figure 1F1. – Data in text indicate constriction of 3.6 +/- 1.3%, but average bar appears at -5% for t-ACPD. Additionally, figure legend indicates p<0.05, but p-value reported in the text is p=0.05.

5) Figure 1G1. – What is the ATP data compared to? A 0% baseline, CO_2_ response or t-ACPD? Additionally, the p-value in the text is denoted as p=0.0018, but is labeled as P<0.001 in the figure legend.

6) Figure 2 and Results. Given how the results are presented in the text (as changes from saline), it is unclear what numbers were used to run the statistics (%ampl or the δ numbers)? If run on the δ numbers, a justification is needed.

7) Based on the% ampl numbers presented in Figure 2, the data are convincing, but the necessity of presenting the numbers differently in the results remains unclear.

8) Why is there only one p-value reported for Figure 2?

9) Response to reviewers indicated additional data had been added to Figure 2, but the U46619 data is not shown in 2B.

10) Figure 3 and Results. Again, the scale for the bar graphs in Figure 3 do not match the numbers presented in the results.

11) The corresponding frequency and amplitude% data are not reported in the text of the results.

*Reviewer #3:*

The authors have performed some additional experiments to tackle the issue of non-specific effects of suramin and PPADS. However, the original concerns remain – the evidence to implicate a purinergic mechanism is based on non-specific pharmacology. I agree the data are suggestive of a purinergic mechanism, but as shown the data do not provide strong evidence to prove this. This weakness also challenges the impact of the observations and the study as a whole for *eLife*.

1) The authors have included the use of 10 μm PPADS in Figure 1 and Figure 3. However, in Figure 1 they still report data with 100 μm suramin. This is perplexing as at this concentration the drug is non-specific.

2) In Figure 3 new data are added with 10 μm PPADS (Figure 3), but the data in Figure 3 are still based on 100 μm PPADS, which is a non-specific concentration. In the last paragraph of the Results and Discussion, they state low concentrations of PPADS are specific, which suggests that they know 100 μm PPADS is non-specific. Why report data with its use at this concentration?

Overall, the manuscript has improved, but the study still relies on high concentrations of drugs that are well known to be non-specific.

Additional experiments with P2X and P2Y receptor agonists and antagonists need to be tried to really prove the mechanism that is proposed. Immunohistochemistry needs to be shown to demonstrate expression of the relevant receptors. Inhibitors of ectoATPases can be tried to explore endogenous ATP involvement. Apyrase could be used to degrade ATP. Ideally, key experiments could be tried in a knock-out mouse of the relevant receptor.

In short, the central finding to implicate ATP is not convincing and more work is needed. The current level of proof (suramin, ATP and PPADS) is where the field was 20-30 years ago. Better tools, drugs and mouse models are available. They should be tried.

[Editors’ note: what now follows is the decision letter after the authors submitted for further consideration.]

Thank you for submitting your article "Purinergic regulation of vascular tone in the retrotrapezoid nucleus is specialized to support the drive to breathe" for consideration by *eLife*. Your article has been favorably evaluated by a Senior Editor and three reviewers, one of whom, Jan-Marino Ramirez (Reviewer #1), is a member of our Board of Reviewing Editors. The following individual involved in review of your submission has agreed to reveal their identity: Jerome Dempsey (Reviewer #3).

The reviewers have discussed the reviews with one another and the Reviewing Editor has drafted this decision to help you prepare a revised submission.

We congratulate you to your discovery of a differential control of the vascular system within the RTN, the region critical for the sensing of CO_2_. It is well established that in response to increased CO_2_/H^+^ the brain, and body in general, responds with vasodilation, a physiological response necessary to increase blood flow and to facilitate oxygenation and removal of CO_2_/H^+^. Your study convincingly demonstrates that the arterioles in the RTN constrict in response to CO_2_/H^+^, which is mediated by purinergic receptors. The RTN is the presumed site for chemoreception and it seems advantageous that the RTN has a differential vasomotor response compared to other sites in the body, a response that seems to be adapted to allow optimal chemoreception. Whether the RTN is the only central nervous system site to differentially control the vasomotor response cannot be deduced from these experiments. But, the study supports the notion that the vascular response is differentially regulated to adapt the blood supply to the specific function of a given CNS region.

We have a few discussion points listed under "Essential revisions" that we would like you to address to further improve the impact of your study.

Essential revisions:

1) We would like you to discuss other studies and observations that are potentially consistent with a local control of vascular reactivity within the brain. We believe that the following studies could be relevant for the concept that you propose in the present study.

A) In healthy humans indomethacin administration reduces the CBF and "cerebrovascular" response to hyper-/hypocapnia in the middle cer artery and many other brain and brain stem regions (see Xie et al. 2006, 2009 ref and references) with corresponding increases in eupneic ventilation and the ventilatory response slope to hyperoxic hypercapnia and hypocapnia.

B) Similar increases in the CO_2_ ventilatory response occurred with carotid occlusion in the goat (Parisi, 1992, Chapman, 1979).

C) Clinically, in post-transplantation anaemia and associated congestive heart failure, cerebrovascular responses to CO_2_ have steep ventilatory responses to CO_2_ contributing to breathing instability and apneas.

Could this (C) and the other findings (A, B) mean that under certain conditions that influence cerebral vascular reactivity "globally", there are marked influences on chemoreceptor responsiveness and ventilatory control stability, because they influence the PCO2 locally at the level of the chemoreceptors? Would these observations be compatible with a qualitatively different vascular responsiveness to CO_2_/H^+^ at the level of the central chemoreceptors as your data supports? A discussion like this could enhance the clinical impact of your interesting finding.

2) We would like you to discuss/clarify the following consideration: Your contention that the unique RTN vascular responsiveness to CO_2_/H^+^ "enhances chemo function","…contributes to the drive to breathe", "…supports neural function and behavior" seems to imply that you believe this specialized function to be beneficial to the organism. However the function of *both* the ventilatory response *and* (at least most of) the cerebral vascular response to hyper/hypocapnia is to minimize the disturbance in CO_2_/H^+^.….so would *not* having the vasodilation/vasoconstriction responses in the RTN mean that the burden is now completely on the (more expensive?) ventilatory response to regulate local H^+^?

3) It would help to include n values in the figures and/or figure legends.

4) The use of "con" and "control" is still confusing. The reviewer thanks the authors for clarifying "con" in the response to reviewers. It would help the readers of the manuscript to benefit from such clarification. This reviewer recommends labeling it as baseline, instead of "con" or "control". Particularly, since the CO_2_ group becomes the control group in comparisons.

5) Figure 1. The figure legend describes that 1B shows the response in PPADs, but only "con" and CO_2_" are shown.

6) Results/Discussion, "under control conditions", "baseline" would be more informative.

7) Figure 3: Like for Figure 1, it would be more informative to refer to "baseline" and "CO_2_" rather than "control conditions".

8) Excellent work from Brian MacVicar's group has gone a long way to helping understand the role of astrocytes in cerebral blood flow, yet reference to this group is only made in the Methods section. Additional acknowledgement to their body of work is needed, including their most recent paper (see below).

9) Results/Discussion, first paragraph: N=34 – is this n=34 animals or 34 vessels? Please clarify.

10) Results/Discussion, second paragraph: Did the authors also do immuno for P2X receptors? Are they expressed there, but not involved? If the authors are eliminating their involvement, it would be useful to know whether or not they are expressed there.

11) Results/Discussion, fifth paragraph: MRS2179 data not shown in figures. Please indicate in the text.

12) Also, a reference to a recent study from MacVicar's group needs to indicate that more is known about dilation in the hippocampal-neocortical/barrel cortex (Howarth et al. 2017 J Neurosci).

13) It would be helpful for the authors to address why pharmacologically manipulating RTN vessels changes amplitude and not frequency in Figure 4, but changes both in Figure 5.

14) Figure 1: Figure 1. The figure legend describes that 1B shows the response in PPADs, but only "con" and CO_2_" are shown.

15) Figure 2: Parts C and D are not addressed in the figure legend.

16) Figure 5: Saline should be open/white bars, not black bars.

17) Methods – your use of "peak" di EMG to quantify "resp muscle activity" is partly dependent on breath timing changes alone. Wouldn't a more appropriate metric be either mean amplitude (total area over TI) or rate of rise of di EMG?

18) You chose to test the "response to hypercapnia" in vivo by raising FetCO2 to 10% or almost twice the air br value in the rodent and showing that the vessel diam changed 3-4%. Given the highly sensitive response of CBF to CO_2_ in the cer vasc (3-4% per mmHg δ PCO2) did you try more physiologic perturbations to show the sensitivity of your prep? Also, given the absence of protein buffers in the cerebral ECF your pH in the RTN is probably in the 6-6.5 range with your level of hypercapnia.

---

## [Author Response]

[Editors’ note: the author responses to the first round of peer review follow.]

*[…] In this study, the role of purinergic modulation of vascular tone is primarily explored by pharmacological manipulations, as is the in vivo demonstration. The use of phenylephrine for example will affect vascular tone, but this manipulation might also have other indirect effects. Given the novelty of this observation, I am not too concerned that most mechanistic insights are based on pharmacological manipulations. However, being primarily a pharmacological study the authors cannot definitely demonstrate e.g. that it is the asctrocytic ATP which antagonizes the CO_2_/H^+^ vasodilation.*

We agree that our pharmacological evidence does not definitively identify astrocytes as the source of CO_2_/H^+^-evoked ATP. Considering that previous work has already shown that RTN astrocytes release ATP when activated by CO_2_/H^+^ or by optogenetic simulation (PMID: 20647426), we have modified the focus of this study to understand how CO2/H^+^-evoked ATP release contributes to respiratory drive by regulating local vascular tone. Specifically, we hypothesize that purinergic signaling provides specialized control of vascular tone by preventing CO_2_/H^+^-induced vasodilation within the RTN. We believe this is an important and broadly relevant focus since virtually nothing is known regarding regulation of vascular tone in any respiratory control center, and because it is not clear whether vascular control might be tailored to support specific neural functions.

We also appreciate your concern that the vasoconstrictor used in vivo (phenylephrine) may have indirect effects. Therefore, we performed additional in vivo experiments using a second commonly used vasoconstrictor (U46619; a thromboxane A2 receptor agonist) (PMID: 15499024; PMID: 17013381). Consistent with our phenylephrine results, we found that application of U46619 to the RTN region increased the ventilatory response to CO_2_ but with no change in mean arterial pressure. These new results have been added to Figure 2. We also used U46619 for our in vitro experiments and can confirm that this drug is a potent vasoconstrictor. Therefore, these results provide strong support for our hypothesis that regulation of vascular tone in the RTN is important functional for control of respiratory behavior.

*Reviewer #2:*

*This is a potentially interesting paper that makes the case that ATP signaling mediates a specialized form of signaling in the brain areas that control breathing. Overall, I found this finding to be a useful contribution, but was not convinced that it was sufficiently novel for a general audience. In some ways, it seems appropriate for a specialized audience of researchers working on breathing.*

We thank the reviewer for their helpful suggestions. We respectfully disagree with the reviewer that the significance of our study is limited to the field of respiratory control. Our discovery that the vascular response to CO_2_/H^+^ in the RTN is opposite to the rest of the cerebrovascular tree is the first evidence to suggest that regulation of vascular tone in any brain region is tailored to support local neural network function and ultimately behavior. The behavior we focus on is breathing, however, the significance of this discovery may be relevant to a wide range of behaviors and so we believe this study will be broadly appealing to any interested in neurovascular control.

*My major concern for the paper is the use of 100 μm suramin and PPADS to implicate purinergic signaling. At these concentrations, neither drug is specific and their ability to block many other receptors is well described in published work. Moreover, much better blockers are available that target distinct ATP receptors. My suggestion is to repeat the key experiments with more selective antagonists and/or to repeat the experiments with suramin and PPADS at 3-10 uM, at which doses they are considered more selective for ATP receptors over receptors for other neurotransmitters. Without a more complete evaluation of ATP's involvement, the central message of this paper appears to be on weak foundations.*

We performed three new sets of experiments to support our conclusion that purinergic signaling prevents CO_2_/H^+^-induced dilation of RTN arterioles. First, as suggested by the reviewer we re- tested the effects of CO_2_/H^+^ on vascular tone in vitro and in vivo using low concentrations of PPADS. We found that 5 µM PPADS was sufficient to eliminate CO_2_/H^+^-induced vasoconstriction of RTN arterioles in vitro. These new data are illustrated in Figure 1C1.

Likewise, in the whole animal we found that application of 10 µM PPADS onto the ventral surface converted CO_2_/H^+^ constriction into a significant dilation. Conversely, in the cortex increasing end expiratory CO_2_ resulted in vasodilation under control conditions and in the presence of low or high PPADS. These new results are shown in Figure 3. These results are entirely consistent with our previous results using higher concentrations of P2 receptor blockers, thus further supporting our main conclusion that CO_2_/H^+^-evoked ATP release causes vasoconstriction RTN arterioles in vitro and in vivo.

Second, we mimicked the effects of CO_2_/H^+^ by exogenous application of ATP. In both the slice and whole animal we found that ATP caused vasoconstriction of RTN vessels. These new results are shown in Figures 1G1 and 3A.

Third, since RTN neurons are activated by CO_2_/H^+^ we re-tested the effects of high CO_2_/H^+^ on vascular tone in the presence of tetrodotoxin (TTX) to block neuronal action potentials. As expected, we found in vitro that CO_2_/H^+^-mediated vasoconstriction was retained in the presence of TTX. These results have been added to the text. Together, these results provide strong evidence for the possibility that CO_2_/H^+^-evoked ATP release (most likely from astrocytes) causes vasoconstriction of RTN arterioles, and in conjunction with our functional in vivo evidence showing that vasoconstriction and dilation of RTN vessels increases and decreases the ventilatory response to CO_2_, show that regulation of vascular tone in the RTN is an important determinant of respiratory behavior.

*Finally, some of the traces in Figure 1 showing changes in blood vessel diameter over time were not particularly clear or convincing (e.g. Figure 1D1, A2, E2). Perhaps bolster these experiments with another in vitro approach to be sure the metrics used are reliable.*

We appreciate the reviewers point that the observed effects of CO_2_/H^+^ and t-ACPD on vessel diameter are modest. However, it is important to recognize that blood flow is dependent on the 4^th^ power of radius; therefore, even small changes in vessel diameter could have profound effects on blood flow. In any case, to bolster our conclusion that purinergic signaling causes vasoconstriction, we confirmed that exogenous ATP also results in vasoconstriction both in vitro. We also recapitulated the effects of CO_2_, PPADS, and ATP on RTN vessels in vivo, thus giving us confidence that what we observed in the slice is real and of functional relevance to respiratory activity.

*Reviewer #3:*

*This is manuscript begins to address the intriguing idea that vascular tone in the chemosensitive area, the retrotrapezoid nucleus (RTN) responds differently to increasing CO_2_/H^+^, and thus, likely contributing to the chemosensitivity response. The authors used both in vitro and in vivo preparations to examine changes in arteriole diameter on the ventrolateral medullary surface during increased CO_2_/H^+^ and after blocking purinergic receptors with suramin or PPADS. Cumulatively, the authors suggest that their results demonstrate that the arterioles near the RTN respond to ATP released from astrocytes to prevent vasodilation in response to increasing CO_2_/H^+^. This is a very interesting and important idea; however, two major issues fail to be addressed in this manuscript. Firstly, the authors also fail to address why different purinergic receptor antagonists are used in the in vitro (suramin) and in vivo (PPADS) experiments.*

We have remedied this issue by testing the effects of PPADS on vascular CO_2_/H^+^ reactivity in both preparations. We found that PPADS prevented CO_2_/H^+^-induced vasoconstriction in vitro and in vivo. Interestingly, in the whole animal RTN application of PPADS converted CO_2_/H^+^- induced vasoconstriction into a significant dilation. This response suggests that in the absence of purinergic signaling RTN vessels respond to CO_2_/H^+^ in a manner similar to the cortex. These new results are shown in Figures 1C1 and 3A-C and the text has been modified accordingly.

*Secondly, the authors fail to acknowledge that suramin, and to a lesser extent PPADS, are associated with non-specific effects, particularly at glutamate receptors in the concentrations used here (for example, see Gu et al., 1998, Neuroscience Letters; Motin and Bennett, 1995, Br. J. Pharmacol.; Nakazawa et al., 1995, Naunyn Schmiedebergs Arch. Pharmacol; Lambrecht, 2000, Naunyn Schmiedebergs Arch. Pharmacol). These two issues at a minimum need to be addressed in the Discussion. In addition, the authors need to address how these non-specific actions could alter their conclusions.*

As noted above we addressed potential off-target effects of our pharmacological approach by performing several sets of experiments. First, we re-tested CO_2_/H^+^-vascular responses in the presence of PPADS at a concentration that reportedly is specific for P2 receptors (PMID: 15351302). We found that 5 µM PPADS blocked CO_2_/H^+^-mediated vasoconstriction of RTN in vitro and 10 µM PPADS blocked CO_2_/H^+^ vasoconstriction of RTN vessels in vivo. These new results have been added to Figures1C1 and 3A, C.

Second, we mimicked the effects of CO_2_/H^+^ by exogenous application of ATP. In both the slice and whole animal we found that ATP caused vasoconstriction of RTN vessels. These new results are shown in Figures 1G1 and 3A.

Third, to minimize a potential neuronal contribution we re-tested vascular responses to CO_2_/H^+^ in vitro in the presence of TTX to block action potentials. We found that CO_2_/H^+^-mediated vasoconstriction was retained in the presence of TTX. These results have been added to the text. Based on these findings we are confident that CO_2_/H^+^-evoked ATP release causes vasoconstriction of RTN arterioles in a manner that supports the drive to breathe. Nevertheless, we added a statement to the last paragraph of the discussion to acknowledge the limitation of our pharmacological approach.

*Lastly, the authors state clearly in the Introduction that they hypothesize "CO_2_/H^+^-evoked astrocyte ATP release will antagonize CO_2_/H^+^-vasodilation in the RTN…"; however, this is not actually the hypothesis that the authors test since they do not directly test the involvement of astrocytes. While this is a logical conclusion based on previous work and in conjunction with the studies tested here, the hypothesis stated needs to accurately describe the experiments herein. In other words, the hypothesis tested is that ATP released during increased CO_2_/H^+^ does not cause vasodilation in the RTN.*

Good point. Since there is compelling evidence showing that RTN astrocytes release ATP when activated by CO_2_/H^+^ or optogenetic simulation (PMID: 20647426), we have modified the focus of this study to understand how CO_2_/H^+^-evoked ATP release contributes to respiratory drive by regulating local vascular tone. As suggested, our new hypothesis is that purinergic signaling provides specialized control of vascular tone by preventing CO_2_/H^+^-induced vasodilation within the RTN. As noted above, this is an important and broadly relevant focus since virtually nothing is known regarding regulation of vascular tone in any respiratory control center, and because it is not clear whether vascular control might be tailored to support specific neural functions.

*Despite these issues, the data presented is very interesting and has the potential to foster additional studies to examine the interaction between astrocytes and the vasculature in this area, similar to work in the cortex.*

[Editors’ note: the author responses to the second round of peer review follow.]

*[…] Reviewer #2:*

*The manuscript by Hawkins et al. is vastly improved by the revisions. The authors addressed the concerns brought forth by the reviewers. And this reviewer remains interested in the results; however, a few lingering issues remain that decrease the clarity of the manuscript. Mainly, some of the results of the current experiments continue to be overstated (see below). As discussed in the previous review, while these conclusions are plausible given previously published results, the results from the present experiments do not directly lead to such conclusions. Wording adjustments are necessary to make clear what is actually shown in the current paper. Secondly, inconsistencies between the data reported in the text of the results and the figures (specifically the bar graphs) is concerning (see below for details). These discrepancies make it challenging to accurately evaluate the manuscript. Overall, this manuscript remains of interest to this reviewer with a few more revisions.*

We thank the reviewer for their helpful suggestions. Based on your suggestions we have toned-down our conclusions and made clear the limitations of our approach. We have also carefully edited the text and figures to ensure all results are presented in a clear and consistent manner.

*1) Concluding sentence overstatements*

*"Together, these results suggest that astrocytes and purinergic signaling have fundamentally different roles in the regulation of vascular tone in the RTN and cortex, with the RTN being specialized to support chemoreceptor function." The reviewer would agree with the conclusion about purinergic signaling, but these data do not directly address involvement of astrocytes. The authors measure arterioles, not astrocyte activity.*

We appreciate the reviewers point but it should also be recognized that t-ACPD is pharmacological tool commonly used to activate astrocytes. We have modified the statement in question to read “These results are consistent with our hypothesis that purinergic signaling, possibly from CO_2_/H^+^-sensitive RTN astrocytes (Gourine et al., 2010), serves to maintain arteriole tone in the RTN during hypercapnia”.

*"These results provide clear evidence that increasing or decreasing vascular tone in the RTN will influence the response to CO_2_. Therefore, we conclude that regulation of vascular tone in the RTN contributes to the chemoreflex." The conclusion for these results is more appropriately worded in the response to reviewers. Since the authors did not verify vascular tone in the experiments related to this conclusion, this conclusion is overstated for what is shown. Despite the addition of the U46619 data, the data does not indicate these drugs ONLY altered blood vessels. Based on the data presented, they cannot rule out diffusion of the drugs and altering astrocytes or neurons in VLM.*

Good point. We have modified the end of that paragraph to read “These results are consistent with the possibility that regulation of vascular tone in the RTN can influence respiratory output. […] It remains to be determined whether U46619 or SNP also affect activity of neurons or astrocytes in the RTN”.

*2) Data inconsistencies*

*Figure 1 and Results. Overall, it is unclear which statistical tests were run and which data were involved in which tests. The Methods indicate both t-tests and ANOVAs were run, but it is not clear when each was run or how data were grouped to determine significance.*

Modified the Methods data analysis section to clarify how the data was analyzed and illustrated. We also added details regarding each statistical test to all figure legends

*3) It is unclear what "control" is for Figure 1. Is control the response to CO_2_ for all data in Figure 1? In specific, is "con" referring to the CO_2_ response or the t-ACPD in Figure 1E1? The same question for Figure 1F2. If in both cases, "con" refers to the CO_2_ response and if that is included when running the statistics, why does this bar not appear in the summary data?*

CO_2_ data in Figure 1.

In Figure 1, the “con” trace shows vessel diameter before exposure to CO_2_ whereas the “CO_2_” trace shows vessel diameter during high CO_2_. However, the summary data shown in Figure 1 is plotted as CO_2_-induced change in vessel diameter so control represents the CO_2_ response in the absence of any blockers and the other bars represent CO_2_ responses in the presence of the specified blocker. To make this clearer we now label the control trace as CO_2_.

To determine if CO_2_/H^+^ significantly affected baseline vascular tone under each experimental condition, we compared the CO_2_/H^+^-induced change in diameter (µm) from baseline alone and in the presence of each blocker by RM-one-way ANOVA and Fishers LSD test and differences were identified by hash marks. We also compared the differences in CO_2_/H^+^-induced% change under control conditions vs. in the presence of blockers by one-way ANOVA and Fishers LSD test and these differences were identified by asterisks.

t-ACPD data in the new Figure 2.

To simplify this figure we moved the t-ACPD for the RTN and cortex into a separate new Figure 2. Again the “con” traces in Figure 2 show vessel diameter before exposure to t-ACPD whereas the “t-ACPD” trace shows vessel diameter during exposure to t-ACPD. The summary data shown in Figure 2 is plotted as t-ACPD induced change in vessel diameter and we now label the control trace as t-ACPD.

To determine if t-ACPD significantly affected baseline vascular tone under control conditions or in the presence of PPADS, we compared vessel diameter (µm) under baseline conditions and in t-ACPD alone or t-ACPD plus PPADS by RM-one-way ANOVA and Fishers LSD test and differences were identified by hash marks. We also compared the difference in t-ACPD-induced

% change under control conditions vs in PPADS by paired t-test.

4) Figure 1F1. Data in text indicate constriction of 3.6 +/- 1.3%, but average bar appears at -5% for t-ACPD. Additionally, figure legend indicates p<0.05, but p-value reported in the text is p=0.05.

We have edited the text to make sure all results are consistent with figures. Note that absolute p values are reported in the text; however, to simplify figures we illustrate significance according to the following: one symbol = p < 0.05, two symbols = p < 0.01, three symbols = p < 0.001, four symbols = p < 0.0001.

5) Figure 1G1. What is the ATP data compared to? A 0% baseline, CO_2_ response or t-ACPD? Additionally, the p-value in the text is denoted as p=0.0018, but is labeled as P<0.001 in the figure legend.

We compared the effects of ATP or other agonists on basal vessel diameter (µm) by RM-one- way ANOVA and Fishers LSD test and differences were identified by hash marks. We also compared ATP-induced% change in diameter to the% change elicited by more specific agonists and these differences were identified by asterisks. We have edited the text and figures to make sure they are consistent.

*6) Figure 2 and Results. Given how the results are presented in the text (as changes from saline), it is unclear what numbers were used to run the statistics (%ampl or the δ numbers)? If run on the δ numbers, a justification is needed.*

We used the percent change from saline to calculate the statistics.

*7) Based on the% ampl numbers presented in Figure 2, the data are convincing, but the necessity of presenting the numbers differently in the results remains unclear.*

Sorry for the confusion. We have modified how the data is presented in the text to match each figure. In this case, the effects of SNP, Phe and U46619 on DiaEMG are presented as percent change from saline; however, since these treatments had no effect on MAP those results are presented as absolute values in mm Hg.

*8) Why is there only one p-value reported for Figure 2?*

These data were analyzed using a one-way ANOVA followed by a Bonferroni multiple- comparison test, therefore only one P values is needed. We have added all relevant p values and specified each statistical test to make clear how the data was analyzed.

*9) Response to reviewers indicated additional data had been added to Figure 2, but the U46619 data is not shown in 2B.*

We have modified this figure (new Figure 4) to include Phe, SNP and U46619.

*10) Figure 3 and Results. Again, the scale for the bar graphs in Figure 3 do not match the numbers presented in the results.*

We have corrected text to make sure all numbers are consistent with figures.

*11) The corresponding frequency and amplitude% data are not reported in the text of the results.*

In the present version of the manuscript we incorporated the% values as suggested by the reviewer.

*Reviewer #3:*

*The authors have performed some additional experiments to tackle the issue of non-specific effects of suramin and PPADS. However, the original concerns remain – the evidence to implicate a purinergic mechanism is based on non-specific pharmacology. I agree the data are suggestive of a purinergic mechanism, but as shown the data do not provide strong evidence to prove this. This weakness also challenges the impact of the observations and the study as a whole for eLife.*

Based on your suggestions, we have performed several additional experiments to support our conclusion that purinergic signaling contributes to regulation of vascular tone in the RTN during exposure to high CO_2_. In short, we i) repeated all key experiments using a lower and more specific concentration of PPADS; ii) performed additional experiments in vitro and in vivo using selective agonists that target P2X or P2Y2 and P2Y4 receptors; iii) confirmed by immunohistochemistry that candidate P2 receptors responsible for CO_2_/H^+^-induced vasoconstriction of RTN arterioles (P2Y2 and P2Y4) by expressed by appropriate cells in the region; and iv) performed additional in vitro experiments using an adenosine receptor blocker or an inhibitor of ectoATPase activity to determine if ATP breakdown products contribute to CO_2_/H^+^-sensitivity of RTN arterioles. We hope you agree that these new results provide compelling support of our hypothesis.

*1) The authors have included the use of 10 μm PPADS in Figure 1 and 3. However, in Figure 1 they still report data with 100 μm suramin. This is perplexing as at this concentration the drug is non-specific.*

We have repeated all suramin experiments with a low concentration of PPADS (5 µM) and can confirm that even at this concentration we were able to largely eliminate the response of RTN arterioles to CO_2_/H^+^ and t-ACPD.

*2) In Figure 3 new data are added with 10 μm PPADS (Figure 3), but the data in Figure 3 are still based on 100 μm PPADS, which is a non-specific concentration. In the last paragraph of the Results and Discussion, they state low concentrations of PPADS are specific, which suggests that they know 100 μm PPADS is non-specific. Why report data with its use at this concentration?*

*Overall, the manuscript has improved, but the study still relies on high concentrations of drugs that are well known to be non-specific.*

We know only show data using low (10 µM) PPADS. Again, this low concentration of PPADS effectively blocked the response of RTN pial vessels to CO_2_/H^+^. These new data should mitigate any concern regarding the use of high concentrations of drug that are known to be non-specific.

*Additional experiments with P2X and P2Y receptor agonists and antagonists need to be tried to really prove the mechanism that is proposed. Immunohistochemistry needs to be shown to demonstrate expression of the relevant receptors. Inhibitors of ectoATPases can be tried to explore endogenous ATP involvement. Apyrase could be used to degrade ATP. Ideally, key experiments could be tried in a knock-out mouse of the relevant receptor.*

Good suggestions. Low concentrations of PPADS is most effective at blocking P2X, P2Y2 and P2Y4 receptors (PMID: 9755289), and since low concentrations of PPADS (5 µM) blocked purinergic modulation of cortical and RTN vessels, we consider P2X and P2Y2 and P2Y4 candidate receptors for purinergic modulation of vascular tone in these regions. Therefore, we tested effects of a selective P2Y2 and P2Y4 receptor agonist (UTPγS) and an agonist with high affinity for P2X receptors (α,β-mATP) on diameter of arterioles in the RTN and cortex. In the RTN, we found that UTPγS but not α,β-mATP mimicked the vascular response to CO_2_/H^+^, thus suggesting P2Y2 and/or P2Y4 contribute to this response. We also confirmed by immunohistochemistry that P2Y2 and P2Y4 receptors are expressed by endothelial cells, smooth muscle cells and astrocytes associated with RTN arterioles. Therefore, we consider purinergic signaling, possibly through P2Y2 and/or P2Y4 receptors, as important.

In the cortex, PPADS blunted CO_2_/H^+^-induced vasodilation in vitro and in vivo, suggesting endogenous purinergic signaling promotes vasodilation in this region. However, exogenous application of ATP or UTPγS both resulted in vasoconstriction in vivo. We also detected P2Y2 and P2Y4 immunolabeling of cells (endothelial cells, smooth muscle and astrocytes) associated with cortical arterioles. Therefore, we suspect these receptors are not targeted by endogenous purinergic signaling during high CO_2_, but when activated by exogenous agonists can influence vascular tone. Since we are primarily interested in understanding the role of purinergic signaling in the RTN so we did not further explore the basis of purinergic dilation in the cortex.

As suggested, we also tested for involvement of adenosine signaling in the RTN by bath application of adenosine, the adenosine receptor blocker (8-phenyltheophylline; 8-PT) and by inhibiting ectonucleotidase activity with sodium metatungstate (POM 1). Although adenosine caused a significant vasodilation of RTN arterioles in vitro; neither 8-PT nor POM1 significantly affected CO_2_/H^+^-induced vasoconstriction, suggesting that purinergic metabolites do not limit purinergic induced vasoconstriction of RTN arterioles.

*In short, the central finding to implicate ATP is not convincing and more work is needed. The current level of proof (suramin, ATP and PPADS) is where the field was 20-30 years ago. Better tools, drugs and mouse models are available. They should be tried.*

We hope you agree that the additional results provide convincing support for a role of purinergic signaling in regulation of vascular tone in the RTN. This pharmacological work is essential for informing future targeted transgenic studies in a different species (mouse), however, we believe our results to be highly significant in their own right and worthy of publication in *eLife*.

[Editors' note: the author responses to the re-review follow.]

*Essential revisions:*

1) We would like you to discuss other studies and observations that are potentially consistent with a local control of vascular reactivity within the brain. We believe that the following studies could be relevant for the concept that you propose in the present study.

A) In healthy humans indomethacin administration reduces the CBF and "cerebrovascular" response to hyper-/hypocapnia in the middle cer artery and many other brain and brain stem regions (see Xie et al. 2006, 2009 ref and references) with corresponding increases in eupneic ventilation and the ventilatory response slope to hyperoxic hypercapnia and hypocapnia.

B) Similar increases in the CO_2_ ventilatory response occurred with carotid occlusion in the goat (Parisi, 1992, Chapman, 1979).

*C) Clinically, in post-tranplantation anaemia and associated congestive heart failure, cerebrovascular responses to CO_2_ have steep ventilatory responses to CO_2_ contributing to breathing instability and apneas.*

*Could this (C) and the other findings (A, B) mean that under certain conditions that influence cerebral vascular reactivity "globally", there are marked influences on chemoreceptor responsiveness and ventilatory control stability, because they influence the PCO2 locally at the level of the chemoreceptors? Would these observations be compatible with a qualitatively different vascular responsiveness to CO_2_/H^+^ at the level of the central chemoreceptors as your data supports? A discussion like this could enhance the clinical impact of your interesting finding.*

We believe it is possible, indeed likely, that global manipulations of CO_2_/H^+^ vascular reactivity (point A) and/or cerebral blood flow (points A-B) which have been shown to influence basal breathing and the ventilatory response to CO_2_ could involve the RTN. Recent evidence showed in the hippocampus and cortex that CO_2_/H^+^ increases vascular tone by a cyclooxygenase (COX)-dependent mechanism (PMID: 28137973). This mechanism may also influence vascular tone in the RTN by limiting purinergic-mediated vasoconstriction. For example, we show in vivo that when purinergic receptors in the RTN are blocked, subsequent exposure to CO_2_/H^+^ results in vasodilation (Figure 5), possibly by a COX-dependent mechanism. Therefore, systemic administration of indomethacin (a COX inhibitor) may increase baseline breathing and the ventilatory response to CO_2_ by potentiating CO_2_/H^+^/purinergic-dependent vasoconstriction of RTN arterioles.

Alternatively, it is also possible that decreasing cerebral blood flow globally (indomethacin or occlusion) will cause widespread tissue and arteriole acidification that may be sufficient to recruit other putative chemoreceptor regions (e.g., NTS or raphe) to further increase chemoreceptor drive to the point where breathing may become unstable (point C).

We modified the fifth paragraph of the Results and Discussion to make the point that purinergic signaling may contribute to cortical CO_2_/H^+^ dilation by influencing prostaglandin E2 release by astrocytes. Also, in the context of the RTN, we added the following to seventh paragraph of the Results and Discussion “[…]disruption of CO_2_/H^+^ dilation would be expected to enhance purinergic-dependent vasoconstriction of RTN arterioles, and thus increase baseline breathing and the ventilatory response to CO_2_. […] It should also be noted that global disruption of cerebrovascular CO_2_/H^+^ reactivity as associated with certain pathological states including heart failure and stroke (Yonas et al., 1993; Howard et al., 2001) or by systemic administration of indomethacin can increase chemoreceptor gain to the extent that breathing becomes unstable (Fan et al., 2010, Xie et al., 2009).”

*2) We would like you to discuss/clarify the following consideration: Your contention that the unique RTN vascular responsiveness to CO_2_/H^+^ "enhances chemo function","…contributes to the drive to breathe", "…supports neural function and behavior" seems to imply that you believe this specialized function to be beneficial to the organism. However the function of both the ventilatory response and (at least most of) the cerebral vascular response to hyper/hypocapnia is to minimize the disturbance in CO_2_/H^+^.….so would not having the vasodilation/vasoconstriction responses in the RTN mean that the burden is now completely on the (more expensive?) ventilatory response to regulate local H^+^?*

We agree that at the global level, CO_2_/H^+^-induced vasodilation and increased breathing help maintain stable tissue CO_2_/H^+^ levels within a narrow range that is suitable for life. However, buffering tissue CO_2_/H^+^ by this mechanism at the level of respiratory chemoreceptors may dampen chemoreceptor drive, thus diminishing the contribution of breathing to CO_2_/H^+^ homeostasis. Perhaps a more economical means of maintaining global CO_2_/H^+^ would be to coordinate vascular and respiratory responses while at the same time buffering global tissue CO_2_/H^+^. Consistent with this, the early work by Millhorn et al., 1984 and colleagues showed by measuring pH in the bulk CSF and at the ventral medullary surface in cats that acute changes in arterial CO_2_ caused rapid changes in pH at the medullary surface but slow and smaller responses in the bulk CSF (PMID: 6427871), thus suggesting tissue pH in the region of the RTN is less efficiently pH buffered possibly because CO_2_/H^+^ does not dilate vessels in this region. Our measures of vascular diameter in vitro (Figure 1, Figure 3) and in vivo (Figure 5) confirm this possibility by showing that CO_2_/H^+^ causes vasoconstriction in the RTN but dilation in the cortex. These results suggest that tissue CO_2_/H^+^ levels near the RTN are less buffered by vascular CO_2_/H^+^ reactivity, possibly as a means of supporting the function role of this region in respiratory chemoreception.

We have added the following statement to the Introduction to support the rationale for our hypothesis “Consistent with this, early studies showed that fast breathe by breathe changes in arteriole CO_2_ correspond with changes in pH measured at the ventral medullary surface (Millhorn et al., 1984), suggesting tissue pH in this region is not highly buffered, possibly because blood vessels in this region do not dilate in response to CO_2_/H^+^. Furthermore, […]”

*3) It would help to include n values in the figures and/or figure legends.*

To avoid making the figures overly busy, we have added N values to all figure legends. Note that all N and p values are also reported in the text.

*4) The use of "con" and "control" is still confusing. The reviewer thanks the authors for clarifying "con" in the response to reviewers. It would help the readers of the manuscript to benefit from such clarification. This reviewer recommends labeling it as baseline, instead of "con" or "control". Particularly, since the CO_2_ group becomes the control group in comparisons.*

Good point. We have redefined our control as baseline in the text and legends for Figure 1-3. In the legend of Figure 4–Figure 5 we define saline injection as control rather than baseline.

*5) Figure 1. The figure legend describes that 1B shows the response in PPADs, but only "con" and CO_2_" are shown.*

We have reworded the figure legend to better describe each panel of the figure.

*6) Results/Discussion, "under control conditions", "baseline" would be more informative.*

Done.

*7) Figure 3: Like for Figure 1, it would be more informative to refer to "baseline" and "CO_2_" rather than "control conditions".*

We have made this change for Figure 1–Figure 3.

*8) Excellent work from Brian MacVicar's group has gone a long way to helping understand the role of astrocytes in cerebral blood flow, yet reference to this group is only made in the Methods section. Additional acknowledgement to their body of work is needed, including their most recent paper (see below).*

We agree that Dr. MacVicar’s group has made substantial contributions to understanding the roles of astrocytes in control of cerebral blood flow. Indeed, we have modified our in vitro approach to use a more physiological O_2_ level based on their evidence that astrocyte control of vascular tone is strongly influenced by oxidative stress (PMID: 18971930). We have also added two additional references to this group’s work (PMID: 25818565; PMID: 28137973), and we have discussed how our results might fit with their recent evidence showing that astrocytes mediate CO_2_/H^+^ vascular responses by releasing prostaglandin E2 (PMID: 28137973). For example, when discussing the role of purinergic signaling in CO_2_/H^+^ dilation of cortical arterioles we added the following “Alternatively, arachidonic acid metabolites are also potent regulators of vascular tone (MacVicar and Newman, 2015) and recent evidence showed that CO_2_/H^+^-mediated vasodilation in the cortex and hippocampus involved activation of cyclooxygenase-1 and prostaglandin E2 release by astrocytes (Howarth et al., 2017). […] However, currently the cellular and molecular basis of purinergic dilation in the cortex remains unknown”.

We also acknowledge that in the absence of functional purinergic receptors, CO_2_/H^+^-mediated dilation of RTN vessels in vivo may involve PgE2 release from astrocytes as recently described by the MacVicar group (PMID: 28137973). Specifically we added the following statement to the Discussion “This suggests that in the RTN, purinergic-mediated vasoconstriction is working against a background CO_2_/H^+^ dilation, possibly mediated by a cyclooxygenenase/prostroglandine E2-dependent mechanism as described elsewhere in the brain (Howarth et al., 2017).”

Also, as noted above, we discuss how disruption of COX1/PgE2 in the RTN might contribute to ventilatory responses to systemic administration of indomethacin. Please see our response to concern 1A-C.

*9) Results/Discussion, first paragraph: N=34 – is this n=34 animals or 34 vessels? Please clarify.*

We have clarified this by specifying vessel or animal with each n value. We have also specified in the Methods section the total number of animals used for in vitro and in vivo experiments.

*10) Results/Discussion, second paragraph: Did the authors also do immuno for P2X receptors? Are they expressed there, but not involved? If the authors are eliminating their involvement, it would be useful to know whether or not they are expressed there.*

Our pharmacological evidence identified P2Y2/4 receptors and the substrates responsible for CO_2_/H^+^-dependent purinergic regulation of RTN arterioles. Therefore, we focused our immunohistochemical experiments only on confirming expression of P2Y2 and P2Y4 receptors near RTN and cortical arterioles. Nevertheless, it remains possible that P2X receptors are expressed in the regions of interest and so we do not exclude the possibility that P2X receptors may contribute to some aspect of vascular control in these regions. In the Results and Discussion we state that “It is also possible that other purinergic receptors contribute to regulation of arteriole tone in these regions”.

*11) Results/Discussion, fifth paragraph: MRS2179 data not shown in figures. Please indicate in the text.*

Done.

*12) Also, a reference to a recent study from MacVicar's group needs to indicate that more is known about dilation in the hippocampal-neocortical/barrel cortex (Howarth et al. 2017 J Neurosci).*

We agree. We refer the reviewer to our answer to concern (8) above. In short, we have discussed our results in conjunction with the recent study by Howarth et al., 2017.

*13) It would be helpful for the authors to address why pharmacologically manipulating RTN vessels changes amplitude and not frequency in Figure 4, but changes both in Figure 5.*

To clarify this point, we added the following “Considering that manipulations of vascular tone preferentially affect respiratory amplitude (Figure 4), whereas application of PPADS, which likely disrupts both direct excitatory effects of ATP on RTN neurons (Wenker et al., 2012) as well as indirect effects of ATP on vascular tone (Figure 5), blunts respiratory frequency and amplitude, suggest that purinergic signaling in the RTN might regulate discrete aspects of respiratory output”.

*14) Figure 1: Figure 1. The figure legend describes that 1B shows the response in PPADs, but only "con" and CO_2_" are shown.*

As noted in concern (5), we have reworded the figure legend to better describe each panel of the figure.

*15) Figure 2: Parts C and D are not addressed in the figure legend.*

We have modified the legend of Figure 2 to state “(C) diameter trace of a cortical arteriole and corresponding vessel image with example profile plots (D) show that exposure to tACPD (50 µM) increase cortical arteriole diameter”.

*16) Figure 5: Saline should be open/white bars, not black bars.*

Thanks.

*17) Methods – your use of "peak" di EMG to quantify "resp muscle activity" is partly dependent on breath timing changes alone. Wouldn't a more appropriate metric be either mean amplitude (total area over TI) or rate of rise of di EMG?*

Sorry for the mistake. In fact our analysis of DiaEMG amplitude was on mean amplitude over time (20 breathing cycles). We have corrected in the data analysis and statistics section to read "Respiratory muscle activity was calculated as the mean amplitude of the integrated DiaEMG over 20 respiratory cycles".

*18) You chose to test the "response to hypercapnia"* in vivo *by raising FetCO2 to 10% or almost twice the air br value in the rodent and showing that the vessel diam changed 3-4%. Given the highly sensitive response of CBF to CO_2_ in the cer vasc (3-4% per mmHg δ PCO2) did you try more physiologic perturbations to show the sensitivity of your prep? Also, given the absence of protein buffers in the cerebral ECF your pH in the RTN is probably in the 6-6.5 range with your level of hypercapnia.*

For this initial characterization we chose a CO_2_ stimulus that is well characterized and thus will facilitate comparisons with previous work. However, now that we have established a foundation for comparison, in future work we plan to characterize vascular responses to CO_2_ over a range of stimuli.